# A New Framework of Green Transition of Cultivated Land-Use for the Coordination among the Water-Land-Food-Carbon Nexus in China

Shandong Niu, Xiao Lyu *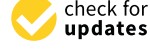 and Guozheng Gu

School of Humanities and Law, Northeastern University, Shenyang 110169, China; 2010005@stu.neu.edu.cn (S.N.); 2010006@stu.neu.edu.cn (G.G.)

\* Correspondence: lvxiao@mail.neu.edu.cn; Tel.: +86-187-6931-9508

**Abstract:** As a fundamental solution to the ecological problems of resources and environment, the Green Transition of Cultivated Land-use (GTCL) has become an inherent requirement for promoting ecological progress and implementing the food security strategy in the new era. This paper proposed a theoretical framework of GTCL and constructed a GTCL development index system based on four aspects: water, land, food and carbon; then, by applying a comprehensive evaluation model, a coupling coordination model and exploratory spatial data analysis, the development level of GTCL in China's 31 provinces, municipalities and autonomous regions in 2000, 2005, 2010, 2015 and 2020 was evaluated and the spatial and temporal rates of change of "water, land, food and carbon" (WLFC) and their coupling coordination were finally analyzed to reveal the "water, land, food and carbon" effect of GTCL. Results showed that the systemic changes of WLFC and its coupling coordination degree of GTCL presented a spatial and temporal coincidence with a high degree of consistency; from 2000 to 2020, the overall GTCL rate in all Chinese provinces, municipalities and autonomous regions showed a "W"-shaped fluctuation uptrend. In the past five years, the development level of GTCL was higher in Northeast China, followed by Central China and North China, while South China was at a low level. In addition, WLFC showed a more obvious "W"-shaped fluctuation, with higher coupling coordination in Northeast China in good coordination and lower coordination in East China and Southwest China. Therefore, according to the results of the study, areas were divided into: benefit leading area, quality improvement area, connotation tapping potential area, ductile development area and ecological reserve area for the regulation of GTCL in all Chinese provinces, municipalities and autonomous regions.

**Keywords:** cultivated land protection; Green Transition of Cultivated Land-use (GTCL); Water-Land-Food-Carbon (WLFC); coupling coordination

## 1. Introduction

Cultivated land serves as an important prerequisite and guarantee for grain yield, and the material basis for human production, life and survival. Since the 21st century, the term "Anthropocene" has been widely mentioned and a series of risk events such as "gray rhinoceros" and "black swan" have evolved into global food crisis, environmental crisis, ecological crisis and major public events, making the whole society more and more concerned about cultivated land conservation. With the further increase of global population, increasingly higher requirements of quality of life, limited growth in cultivated land area, and a series of hazards of environmental pollution to the ecosystem, food security has become an important topic of global sustainable development [1–3]. According to the report "The State of the World's Land and Water Resources for Food and Agriculture in 2021: System at breaking point" released by the Food and Agriculture Organization of the United Nations (FAO), the state of the planet's soil, land and water resources has deteriorated dramatically over the past decade, increasing pressure to meet the food needs

of nearly 10 billion people worldwide by 2050. If agricultural water withdrawals are likely to increase by about 35% in order to achieve a 50% increase in grain yield, the world will face a major challenge in providing water, energy and food. Admittedly, the enormous pressure to increase grain yield also comes with the cost of global ecological degradation. Deforestation for agriculture is estimated to be a significant source of greenhouse gas emissions, accounting for 17% of global carbon emissions, as well as harms such as soil erosion and biodiversity reduction [4]. Moreover, the heavy use of chemical fertilizers, pesticides, and plastic films can lead to significant nitrous oxide emissions and affect global warming [5]; excessive use of nitrogen and phosphorus fertilizers can cause (leaching, erosion) pollution of water resources, and its production can result in $CO_2$ emissions [6]; plastic residual films can damage the soil structure of the tillage layer, causing difficult infiltration of groundwater and secondary soil salinization [7]; pesticides present a risk if they are not used in a proper professional way, and even damage the ecological environment through the atmosphere, water bodies, soil, and food [8]. Therefore, there is an urgent need to find sustainable ways to increase food supply and protect the environment in agroecosystems.

As a traditional agricultural power, China feeds 22% of the world's population with 7% of the world's cultivated land [9]. Its grain yield has remained stable at over 1.3 trillion jin (a metric unit equal to exactly 500 g) for seven consecutive years [10], but there is also a "double high" phenomenon of high stocks and high imports. The rapid development of urbanization has led to the loss of cultivated land [11], and the number in hectares per capita of cultivated land areas is less than 1/2 of the world average; the pursuit of unit grain yield and excessive intensive production have brought serious negative environmental effects, and exposed cultivated land resources to the serious situation of declining soil organic matter, soil erosion, soil acidification, rapid decline of groundwater, and farmland pollution [12,13]. China's cultivated land system is on the verge of breaching its environmental boundary threshold. Relying on a large number of chemical inputs, China consumes nearly a third of the world's chemical fertilizers and uses 2.5 times more pesticides per unit area than the world average [14]. N and P losses in 2018 were 821.5 kilotons and 2137.8 kilotons, respectively, which fall in the high-risk zone [15]. China accounts for 13% of global irrigation water use, has a blue water footprint of 224.5 billion cubic meters and suffers from acute water scarcity [16]. China contributes 33% and 36% of global excess agricultural nitrogen and phosphorus and is responsible for 28% of all global $N_2O$ emissions from croplands [3]. In addition, 19.4% of China's cultivated land is subject to soil contamination, which adversely affects grain yield [17,18]. In the short term, China's food supply is relatively adequate and maintains a tight balance with the ecological environment. However, in the long term, the spatial distribution of China's water, land and food resources is uneven [12] and ensuring food security remains China's top priority. The "North-to-South Grain Transfer" has put the north, where water resources are insufficient, in a dilemma of water resource "overload" and ecological environment degradation, which is not conducive to sustainable socio-economic development. As China's 14th Five-Year Plan suggests, emphasis should be, once again, put on promoting green development, pushing comprehensive green transition of economic and social development, and modernizing harmonious coexistence between man and nature. As stated in Subsection 8, "The action for consolidating and enhancing carbon sink" of the Action Plan for Carbon Dioxide Peaking Before 2030, efforts shall be stepped up to promote carbon emission reduction and carbon sequestration in agriculture and rural areas, develop green, low-carbon and circular agriculture, and take action to improve the quality of cultivated land [19]. Therefore, the coordinated and sustainable development of water resources, land resources, energy (carbon) and food systems is a major issue that the country needs to address urgently.

Food, energy and water are the three basic human needs, and their coupling has become a major global concern [20–22]. The society also faces major challenges in providing water, energy and food [23]. Bonn 2011 Conference: The Water, Energy and Food Security Nexus first proposed using "nexus" to see the complex relationship between the three. In the 17 Sustainable Development Goals (SDGs) set by the United Nations,

SDG 2: Zero Hunger (sustainable food production), SDG 12: Ensure sustainable consumption and production patterns (transition to sustainable development), and SDG 15: Ensure the conservation, restoration and sustainable use of terrestrial ecosystems (protect terrestrial ecosystems and promote the sustainable use of ecosystems) relate to the sustainable use of each element of the WEF respectively. The WEF nexus has become the basis for achieving the SDGs [24]. In China, WEF-nexus-related studies are limited at the inter-regional level, mainly focusing on the safety evaluation and influence factors of WEF [25–27] or on one aspect or "water-land", "land-carbon", "water-carbon" and "food-water" relationships [28,29], which is not sufficient to coordinate the sustainable development of regional agriculture. Theoretically, Zhao [30] proposed a system coupling model of "water-land-energy-carbon" to reflect the matching relationship of various regional resources, the efficiency of exploitation and utilization, and the degree of human and social impacts on the environment, so as to achieve the goals of efficient utilization of regional water-land-energy resources, carbon emission reduction, and social production optimization. Subsequently, it was extended to water-land-food-energy nexus [31], land-water-food-environment nexus [32] or food-water-land-ecosystem nexus [33]. However, food security deserves further attention in practical studies of cultivated land-use.

The above studies show that the WEF nexus, based on broad and multiple perspectives, can make a significant contribution to the field of sustainability. However, there is still a scarcity about WLFC nexus; synergistic water-land-food-carbon development is rarely discussed and the complexity of policy formulation and management regarding GTCL is still in shortage. Therefore, the motivation and novelty of this study is to fill this gap in the WLFC-nexus field and to quantitatively assess the development of GTCL from the perspective of WLFC. The details of the objectives are as follows: 1. to construct a theoretical framework for WLFC-based GTCL; 2. to dissect the spatial and temporal differentiation of GTCL; 3. to explore the coupling coordination of WLFC in cultivated land systems; 4. to propose a ductile regulation strategy based on WLFC nexus.

## 2. Theoretical Framework and Research Methods

### 2.1. Case Study and Data Sources

Given the availability of data, choosing the panel data of 31 provinces, municipalities and autonomous regions of China from 2001 to 2021 as research units, this study treated 31 Chinese provinces, municipalities and autonomous regions. Figure 1 depicts the study area, but for the time being, due to data loss, Hong Kong, Macao and Taiwan are not included. The data in this paper are obtained from the 2001, 2006, 2011, 2016 and 2021 China Statistical Yearbook and China Rural Statistical Yearbook.

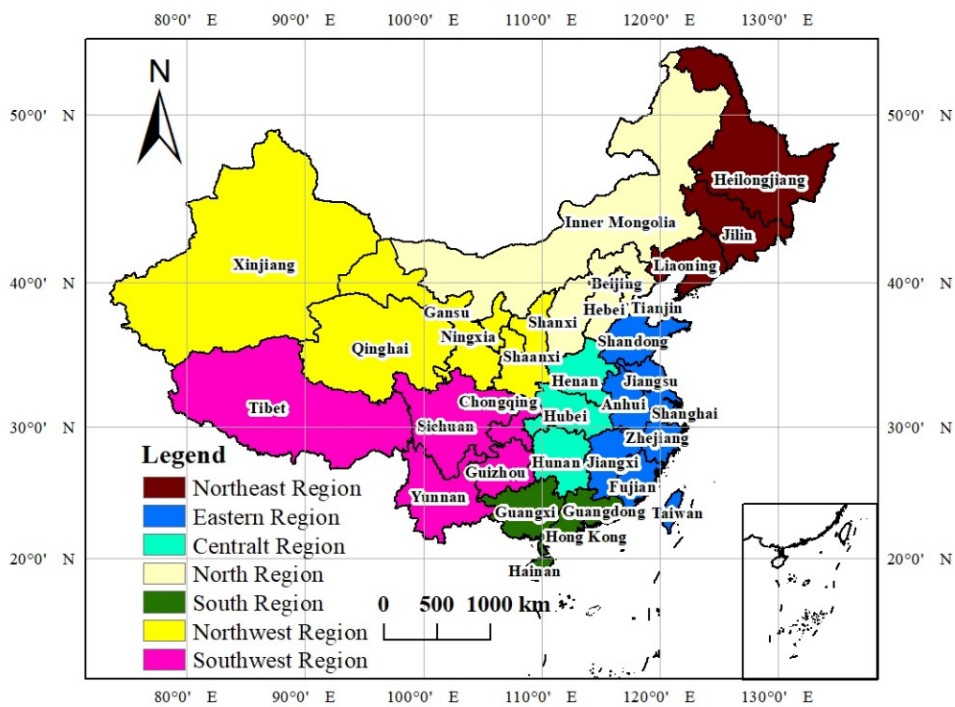

**Figure 1.** Study area.

### 2.2. Green Transition of Cultivated Land-Use (GTCL)

Cultivated land is the land used for crop planting, growing and harvesting as well as agricultural products production. A cultivated land system refers to the natural-ecological system of cultivated land resources and the economic-social system of human activities around the conservation and sustainable use of cultivated land resources in a specific regional space and in a certain time sequence and the complex system formed by the interaction of "element-structure-function-value" transition process through trade-offs/synergies, antagonism/adaptation, gain/loss, etc. [34]. Through the exchange of biotic and abiotic elements with other systems, material circulation, energy flow and information transfer, etc. are constantly taking place within the system. A cultivated land system has a whole development life cycle featuring element inputs, planting management, crop production, grain outputs, etc. Water, land, food and carbon are not only key elements of cultivated systems, but also important resource bases for natural and socio-economic life [35]. "Water" is the material and medium necessary for life activities and the raw material for most industrial production processes; the water system in a cultivated land system includes surface water, groundwater and unconventional water supply [36]. "Land" is the basis and place of human production activities and the source of nutrients for crop growth. Land is an important and non-renewable scarce resource in the cultivated land system, and socio-economic development must depend on land resources, which not only store energy and water resources, but also carry human living space and food needed for life. "Food" is the power source of economic and social systems, and the "food" system in the cultivated land system includes the supply and demand of food [37]; "carbon" is the material basis of living organisms and major energy sources, as well as the emission and metabolites of various human socio-economic activities [38]; carbon emissions from cultivated land systems come from various direct or indirect carbon emissions during the life cycle of cultivated land-use, mainly from pesticides, chemical fertilizers, agricultural films, tillage, agricultural machinery and irrigation. The development and utilization of water, land, food and carbon resources are interrelated and interact with each other. Since the regional cultivated land-use system is complex and has a large spatial heterogeneity, water-land-food combinations and carbon emission intensity vary with natural and social zones and industries in the region.

As shown in Figure 2, based on the life cycle development process of the cultivated land-use system, elements of different types in the "nature-culture" involving water, land, food, and carbon form a new subsystem in the "horizontal-vertical" structure through the material cycle, energy flow and information transfer. However, there is no linear evolutionary substitution between the four subsystems of "water-land-food-carbon" and the interaction between them is enhanced along with the human demand for the diversification, hierarchy and regionalization of cultivated land, as well as better understanding of the versatility and value (physical quantity) assessment of cultivated land [39,40]. For example, the coupling and mutual feedback, spatial and temporal evolution and functional transition of these subsystem components give multiple value attributes to cultivated land resources while forming a fluctuating and self-organized transition in the process of "material-energy-information" exchange/reorganization, "production-ecology-life" functional trade-offs/cooperation and "economy-ecology-society" service value response. Specifically, in the "one-to-one" factor coupling relationship, based on climate, topography, soil conditions, hydrology and biology, the "horizontal structure" and "vertical structure" of the cultivated land-use system are used to find the optimal vertical combination of water, soil, climate and biology and the best suitable interval for crop growth and soil synthesis, forming a sequence of inputs/outputs such as solar radiation, labor, agricultural technology, capital and primary agricultural products, which mainly reflects the spatial pattern of green transition of cultivated land-use. In the "one-to-many" factor coupling relationship, it is mainly the regulation, buffering and self-adaptation of the cultivated land-use system and the external artificial control environment, and different cultivated land-use methods, intensity of use and structural characteristics, with a specific spatial and temporal order and factor ratios to generate a hierarchical composite structure to maintain the material and energy flow exchange, product and value exchange with the external environment. This in turn affects the landscape pattern, soil environment, water environment, air environment, agricultural inputs (fertilizer and pesticide application, irrigation inputs), mechanization level, multiple crop index, crop-soil relationship and cultivated land output, which mainly reflects the evolution of GTCL. In the "many-to-many" factor coupling relationship, throughout the key nodes of cultivated land cultivation, quality improvement, crop maintenance, soil degradation, ecological management, etc., with reference to the source reduction and regulation, process precision and intelligent control, and agile management at the end, focusing on the value of the virtuous cycle of the cultivated land ecosystem, in the process of resource sharing, structural remodeling, functional gain and value response of the cultivated land utilization system, based on the reduced and harmless input/output model, the intensity of material consumption is effectively reduced, soil organic quality is improved, groundwater depletion is slowed down, the intensity of non-point source pollution is reduced and greenhouse gas emissions are reduced, which mainly reflects the system services of GTCL. Throughout the overall process of GTCL, "water" is the "lifeblood", "land" is the "root", "food" is the "core" and "carbon" is the "service".

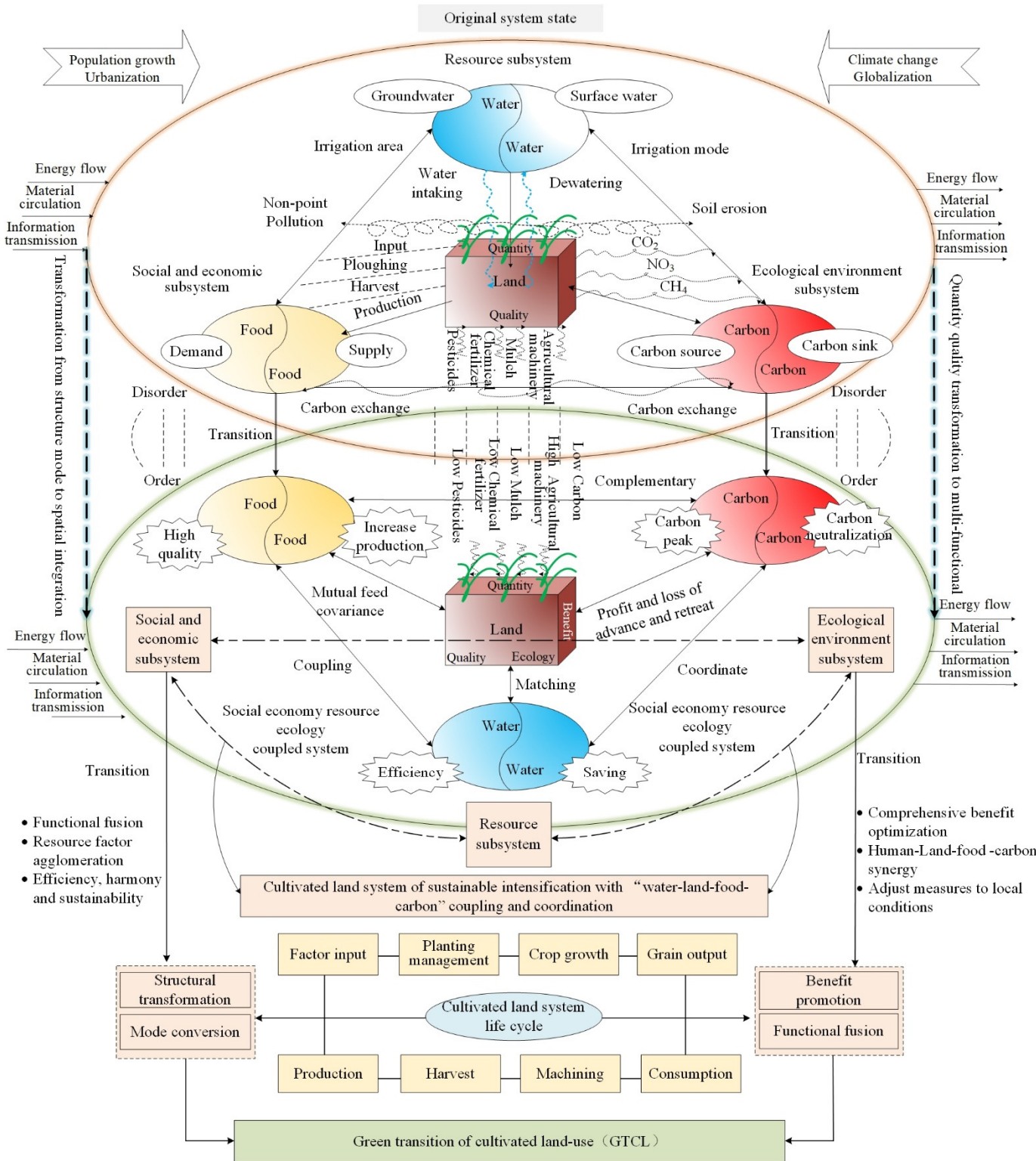

**Figure 2.** The concept framework of GTCL.

## 2.3. Evaluation Index System of GTCL

Based on the connotation of GTCL (Figure 2), this paper constructs an evaluation index system of GTCL, including four factor layers and 20 index layers, and the specific indexes are explained in Table 1.

Table 1. Index system of GTCL.

| Factor Layers | Index Layers | Unit | Weight | Direction | Index Interpretation |
|---|---|---|---|---|---|
| Water | Virtual water self-sufficiency rate | % | 0.129 | + | Internal self-sufficiency of virtual water/total consumption of virtual water, it reflects the self-sufficiency of food production and the contribution of local water resources in residents' food consumption. |
| | Effective irrigation area | hm$^2$ | 0.065 | + | Water source environment reflecting cultivated land resources. |
| | Virtual water land density | m$^3$·hm$^2$ | 0.098 | + | Internal self-sufficiency of virtual water/Grain sowing area, it reflects the water consumption of grain production per unit area. |
| | Water consumption per unit area | L/hm$^2$ | 0.012 | - | Agricultural irrigation water consumption/cultivated land area, it reflects the actual water consumption per unit cultivated land area. |
| Land | Per capita cultivated land area | hm$^2$/person | 0.138 | + | Cultivated land area/total population, it reflects the change of cultivated land quantity. |
| | Land reclamation rate | % | 0.035 | + | Cultivated land area/total land area, it reflects the degree of cultivated land development. |
| | Multiple crop index | % | 0.033 | + | Crop sowing area/cultivated land area, it reflects the degree of cultivated land-use. |
| | Investment ratio of saving and increasing | / | 0.096 | + | Total power of agricultural machinery per labor/chemical input per land, that is, labor-saving input/yield increasing input, reflecting the change of cultivated land input structure. |
| | Non-point source pollution intensity | kg/hm$^2$ | 0.010 | - | The total loss of fertilizer nitrogen (phosphorus), pesticide and agricultural film reflects the carrying capacity of cultivated land ecological environment. |
| | Disaster area | hm$^2$ | 0.009 | - | Reflect the resilience of cultivated land system |
| Food | Average grain yield | kg/hm$^2$ | 0.027 | + | Grain crop yield/cultivated land area, reflecting grain production capacity. |
| | Per capita grain yield | person /kg | 0.134 | | Food crop production/total population, reflecting food production security. |
| | Proportion of sown area of grain crops | % | 0.026 | + | Grain crop planting area/cultivated land area, reflecting grain production potential. |
| | Ratio of food crops to cash crops | % | 0.127 | + | Grain crop area/cash crop area, reflecting the change of grain structure. |
| Carbon | Carbon emission from pesticide use | kg/hm$^2$ | 0.013 | - | |
| | Carbon emission from fertilizer use | kg/hm$^2$ | 0.007 | - | |
| | Carbon emission from plastic film use | kg/hm$^2$ | 0.004 | | Carbon emissions from pesticides, chemical fertilizers, agricultural film, tillage, total power of agricultural machinery and irrigation during the life cycle of cultivated land-use. |
| | Carbon emission from tillage | kg/hm$^2$ | 0.015 | - | |
| | Irrigation carbon emission | kg/hm$^2$ | 0.015 | - | |
| | Carbon emission of agricultural machinery | kg/hm$^2$ | 0.008 | - | |

The indexes of the "water" system include the virtual water self-sufficiency rate, effective irrigation area, virtual water land density, and water consumption per unit of cultivated land. Virtual water for major food crops refers to the amount of water required

for grain yield [41]. In this paper, five major food crops are selected: rice, wheat, corn, soybeans and potatoes. The virtual water self-sufficiency rate is calculated as follows:

$$I = \frac{IVW}{TVW} \tag{1}$$

$$TVW_{(j,t)} = IVW_{(j,t)} - EVW_{(j,t)} \tag{2}$$

$$IVW_{(j,t)} = VW_{(j,t)} \times \sum_{i=1}^{n} G_{(j,t)} \tag{3}$$

$$EVW_{(j,t)} = VW_{(j,t)} \times \sum_{t=1}^{n} \Delta A_{(j,t)} \tag{4}$$

In Equations (1)–(4), $I$ is the virtual water self-sufficiency rate, $IVW$ is the internal virtual water self-sufficiency of grain, $TVW$ is the total virtual cultivation land consumption of grain. $EVW_{(j,t)}$ is the external virtual water flow of crop $t$ in the area $j$, $VW_{(j,t)}$ is the virtual water content of crop $t$ in the area $j$, $G_{(j,t)}$ is the grain output of crop $t$ in the area $j$, $\Delta A_{(j,t)}$ is the transport amount of crop $t$ in the area $j$.

The relevant indexes of the "land" system include per capita cultivated land area, land reclamation rate, multiple crop index, investment ratio of saving and increasing, soil organic matter content, non-point source pollution intensity and affected area. The non-point source pollution of cultivated land is mainly caused by the excessive use of chemical fertilizers, pesticides and agricultural films [42]. Therefore, this paper uses the loss of fertilizer nitrogen (phosphorus), ineffective use of pesticides and agricultural film residues to characterize the pollution level. The average fertilizer, pesticide and plastic film pollution intensity are used to estimate the level of agricultural non-point source pollution, and the calculation equation is as follows:

$$E = \sum E_{ij} = \sum C_{ij} \times \eta_{ij} = \sum T_i \times \rho_{ij} \times \eta_j \tag{5}$$

$$EI = E/AL \tag{6}$$

In Equations (5) and (6), $E$ is the non point source pollution intensity of total fertilizers, pesticides and plastic films, $EI$ is the non point source pollution intensity of average fertilizers (kg/hm$^2$); AL is the total sown area (hm$^2$); $\sum E_{ij}$ denotes the total amount of the $j$ th pollutant produced in the area $i$; $C_{ij}$ denotes the total amount of fertilizers, pesticides and plastic films produced by the $j$ th pollutant in the area $i$; $\eta_{ij}$ denotes the loss rate of the $j$ th fertilizer in the area $i$. $T_i$ is the index statistics of area $i$; $\rho_{ij}$ indicates the product coefficient of the $j$ th pollutant in the area $i$. The coefficients of fertilizer loss, pesticide residue and plastic film residue are shown in Table 2.

The relevant indexes of the "food" system include the average grain yield, per capita grain yield, the proportion of sown area of grain crops and the ratio of food crops to cash crops. Among them, oilseeds, cotton, hemp, sugar, tobacco and vegetables are mainly selected as cash crops. The relevant indexes for the "carbon" system include carbon emissions from pesticide use, carbon emissions from fertilizer use, carbon emissions from agricultural film use, carbon emissions from tillage, carbon emissions from irrigation, and carbon emissions from agricultural machinery. The Intergovernmental Panel on Climate Change (IPCC) guidelines were employed to calculate the $CO_2$ emissions for the available energy consumption (Intergovernmental Panel on Climate Change, 2007), emissions from the consumption of fossil fuels (mainly diesel) by agricultural machinery, indirect emissions from the consumption of electricity (mainly thermal power) by irrigation, and the loss of organic carbon due to tillage [31,44]. With reference to previous studies, carbon emissions are mainly from pesticides, fertilizers, plastic films, tillage, agricultural machinery and irrigation during the life cycle of cropland use (E). The equation for measuring carbon emissions from cropland use is:

$$E = \sum E_i = \sum (G_i \times \delta_i) \tag{7}$$

In Equation (7), $E$ is the total carbon emission from cropland use; $E_i$ is the carbon emission from the $i$ th carbon source; $G_i$ is the original amount of each carbon emission source, and $\delta_i$ is the carbon emission coefficient, which are 0.8956 (kg/kg) for fertilizers [45], 4.9341 (kg/kg) for pesticides [46], 5.18 (kg/kg) for plastic films, 0.18 (kg/kW) for agricultural machinery power (kg/kW), 20.476 kg/hm$^2$ for irrigation [47], and 312.6 (kg/km$^2$) for tillage [48].

**Table 2.** List of product factors of pollutants.

| | Region | Loss Rate /% | Loss Rate /% | Region | Residual Rate /% | Loss Rate /% |
|---|---|---|---|---|---|---|
| | | Nitrogenous fertilizer | Phosphate fertilizer | | Plastic film | Pesticide |
| I | Jiangsu, Beijing | 30 | 7 | Inner Mongolia, Shanxi, Heilongjiang, Jilin, Liaoning | 17.3 | 0.13820 |
| II | Tianjin, Guangdong, Zhejiang, Shanghai | 30 | 4 | | 25.75 | 0.00768 |
| III | Hubei, Fujian, Shandong | 20 | 7 | Tianjin, Beijing, Shandong, Hebei, Henan, Jiangsu | 25.65 | 0.06980 |
| IV | Hebei, Shaanxi, Liaoning, Yunnan, Ningxia, Hunan, Jilin, Inner Mongolia, Guizhou | 20 | 4 | Fujian, Guizhou, Hunan, Jiangxi, Yunnan, Sichuan, Chongqing, Guangdong, Guangxi, Hainan | 13.3 | 0.145625 |
| V | Henan, Heilongjiang | 10 | 7 | Anhui, Zhejiang, Hubei, Shanghai | 22.3667 | 0.228531 |
| VI | Anhui, Hainan, Xinjiang, Shanxi, Guangxi, Gansu, Sichuan, Jiangxi, Chongqing, Qinghai, Tibet | 10 | 4 | Shaanxi, Gansu, Ningxia, Qinghai, Xinjiang, Tibet | 34.41667 | 0.00010 |

Note: The correlation coefficient mainly adopts the literature research method and the relevant data published by the National Bureau of statistics, with reference to Lai [43] and the first national pollution survey: Manual of fertilizer loss, pesticide loss and film residue coefficient, and the impact of regional gap shall be considered as much as possible in the accounting process.

*2.4. Determination of Index Weights*

(1) Entropy weight method

This paper applies the entropy weight method to determine the objective weights of each index in Table 1, and reflects the contribution size of the comprehensive index of GTCL as the final weight value of the evaluation index system based on the generated weight structure. The results are shown in Table 1.

The extreme value standardization was used to unify the indexes to [0, 1] in order to eliminate the influence brought about by different index magnitudes.

For positive and negative indexes, the equation below was used:

$$r_{ij} = \frac{X_{ij} - X_{min}}{X_{max} - X_{min}} \quad r_{ij} = \frac{X_{max} - X_{ij}}{X_{max} - X_{min}} \tag{8}$$

Using the entropy weight method for the calculation of index weights, in an evaluation problem with m evaluation indexes and $n$ evaluation objects, the entropy of the $i$ th indicator is defined as:

$$H_i = -k \sum_{j=1}^{n} f_{ij} \ln f_{ij} \quad i = 1, 2, \cdots, m \tag{9}$$

In Equations (8) and (9): $f_{ij} = \frac{r_{ij}}{\sum_{j=1}^{n} r_{ij}}$, $k = \frac{1}{\ln n}$, when $f_{ij} = 0$, make $f_{ij} \ln f_{ij} = 0$.

After defining the entropy of the $i$ th index, the entropy weight of the $i$ th index is:

$$w_i = \frac{1 - H_i}{m - \sum_{i=1}^{m} H_i} \tag{10}$$

In Equation (10): $0 \ll w_i \ll 1$, $\sum_{i=1}^{m} = 1$.

(2) Comprehensive evaluation model

The equation of multidimensional evaluation:

$$F_i = \sum_{j=1}^{n} W_{ij} \times T'_{ij} \tag{11}$$

In Equation (11): $F_i$ is the GTCL index in different dimensions; $W_{ij}$ is the weight of the $i$ th evaluation index; $T'_{ij}$ is the normalized value of the $j$ th of the $i$ index.

(3)　Scientific test of index system

$$R^2 = 0.992 \quad R^2 = \frac{\sum_i^n \left( Y_{pi} - Y_m \right)^2}{\sum_i^n \left( Y_i - Y_m \right)^2}$$

$$e = \sqrt{1 - R^2} \approx 0 \tag{12}$$

In Equation (12): The $R^2$ value is between 0 and 1 to show how close the predicted value $Y_{pi}$ is to the actual value $Y_i$, $Y_m$ denotes the average value, and $n$ denotes the total number. The closer the residual coefficient $e$ is to 0, the more representative and relevant the index system is.

*2.5. Evaluation Methods*

(1)　Coupling coordination model

$$C = \left\{ \frac{U_1 \times U_2 \times U_3 \times U_4}{\left( \frac{U_1 + U_2 + U_3 + U_4}{4} \right)^4} \right\}^{\frac{1}{4}} \tag{13}$$

$$U_i = \sum_{j=1}^{n} w_j y_{ij} \tag{14}$$

In the Equations (13) and (14): $C$ is the coupling degree value, $U_1$, $U_2$, $U_3$, $U_4$ represent the four subsystems, i.e., "water, land, food, carbon," respectively, and the value range is [0, 1]. The interval of $C$ represents the degree of interrelationship of subsystems. The higher $C$ value indicates the stronger interrelationship between the subsystems, presenting the trend of orderly evolution; on the contrary, the evolution between the systems shows no trend.

$$D = (C \times T)^{\frac{1}{2}} \tag{15}$$

$$T = a \times U_1 + b \times U_2 + c \times U_3 + d \times U_4 \tag{16}$$

In Equations (15) and (16): $D$ is the value of coupling coordination; $T$ is the comprehensive index of GTCL, a, b, c and d are the weight coefficients of subsystems. The higher $D$ value indicates the higher degree of coordination of the cultivated land composite system, and the level of coupling coordination is classified in Table 3.

**Table 3.** Classification of coupling coordination degree.

| Coupling Coordination Degree | Coordination Level | Coupling Coordination Degree | Coordination Level |
|---|---|---|---|
| (0.80, 1.00] | Highly coordinated | (0.30, 0.40] | Reluctantly coordinate |
| (0.60, 0.80] | Good coordination | (0.20, 0.30] | Verge of disorder |
| (0.50, 0.60] | Moderate coordination | (0.10, 0.20] | Moderate disorders |
| (0.40, 0.50] | Low coordination | [0, 0.10] | Serious disorders |

Based on relevant literature [41,44] and the actual situation, this paper uses the natural fracture method to cluster the values of "water-land-food-carbon" internal coordination development degree in the study area with the principle of maximum variance between groups and minimum variance within groups, and classifies the coupling coordination

degree into serious disorder ($0 < D \leq 0.10$), moderate disorder ($0.10 < D \leq 0.20$), verge of disorder ($0.20 < D \leq 0.30$), reluctant coordination ($0.30 < D \leq 0.40$), low coordination ($0.40 < D \leq 0.50$), moderate coordination ($0.50 < D \leq 0.60$), good coordination ($0.60 < D \leq 0.80$), and high coordination ($0.80 < D \leq 1.00$) (Table 3), in order to set the internal coordination development discriminatory criteria.

(2)    Exploratory Spatial Data Analysis (ESDA)

Exploratory Spatial Data Analysis (ESDA) is a visual analysis of spatial data interactions based on spatial correlation to explore the potential relationships of data distribution. The global spatial autocorrelation analysis can be applied to examine the spatial clustering characteristics of GTCL based on the Global Moran's *I* index, which reflects the similarity of attribute values of spatial neighboring areas.

$$I = \frac{n}{\sum_{i=1}^{n}\sum_{j=1}^{n} W_{ij}} \times \frac{\sum_{i=1}^{n}\sum_{j=1}^{n} W_{ij}(x_i - \overline{x})(x_j - \overline{x})}{\sum_{i=1}^{n}(x_i - \overline{x})^2} \tag{17}$$

where, $x_i$, $x_j$ are the cultivated land-use transition indexes in areas *i* and *j*, respectively; $\overline{x}$ is the average of GTCL index in each area; $W_{ij}$ is the spatial weight matrix (adjacency of spatial units), but if areas *i* and *j* are adjacent, $W_{ij} = 1$, otherwise $W_{ij} = 0$. Global Moran's *I* index takes values between $-1$ and 1, and there is no spatial autocorrelation when $I = 0$. There is positive correlation when $I = 0$, and negative correlation when $I < 0$.

### 2.6. Uncertainties and Shortcomings

This paper preliminarily reveals the WLFC nexus in the evolution process of GTCL and measures the coupling coordination degree between them on this basis. Due to the inconsistency of statistical caliber of data in different provinces, this current research has not well studied the internal development of each subsystem of WLFC. However, the research data timeline for official statistics in this paper is selected according to the Five-Year Plan, which will be beneficial to provide a theoretical and empirical ground for formulation and implementation of cultivated land protection policy.

### 3. Results
#### 3.1. Spatial and Temporal Patterns of GTCL
3.1.1. Dynamic Evolution Characteristics of Regional Differences

As shown in Figures 3 and 4, the overall GTCL index in all China's provinces, municipalities and autonomous regions shows a "W"-shaped fluctuating uptrend, with the average values of the comprehensive transition index in 2000, 2005, 2010, 2015 and 2020 being 0.202, 0.137, 0.206, 0.147 and 0.237, respectively. The provinces, municipalities and autonomous regions above the average in 2000 were mainly Heilongjiang, Inner Mongolia, Anhui, Henan and Ningxia; in 2005, they were concentrated in Hebei, Jiangsu, Anhui, Shandong, Henan and Chongqing; in 2010, they were mainly in Inner Mongolia, Jilin, Heilongjiang, Anhui, Henan and Ningxia; in 2015, they were concentrated in Heilongjiang, Anhui, Jiangxi, Shandong and Henan; in 2020, they were concentrated in Inner Mongolia, Jilin, Heilongjiang, Jiangsu, Shandong and Henan. From 2000 to 2005, Inner Mongolia had the largest transition rate at $-48.23\%$, Anhui the smallest at $-19.29\%$; from 2005 to 2010, Heilongjiang had the largest transition rate at $169.07\%$ and Beijing the smallest at $17.72\%$; from 2010 to 2015, Ningxia had the largest transition rate at $-46.00\%$ and Shanghai the smallest at $4.44\%$; from 2015–2020, Inner Mongolia had the largest transition rate at $-51.54\%$, and Shanghai the smallest at $-4.23\%$. In addition, the GTCL in provincial areas is more balanced. The extreme difference of GTCL index is 0.157 in 2000, 0.095 in 2005, 0.301 in 2010, 0.158 in 2015, and 0.673 in 2020, which indicates that the difference of GTCL between provincial areas shows an overall increasing trend.

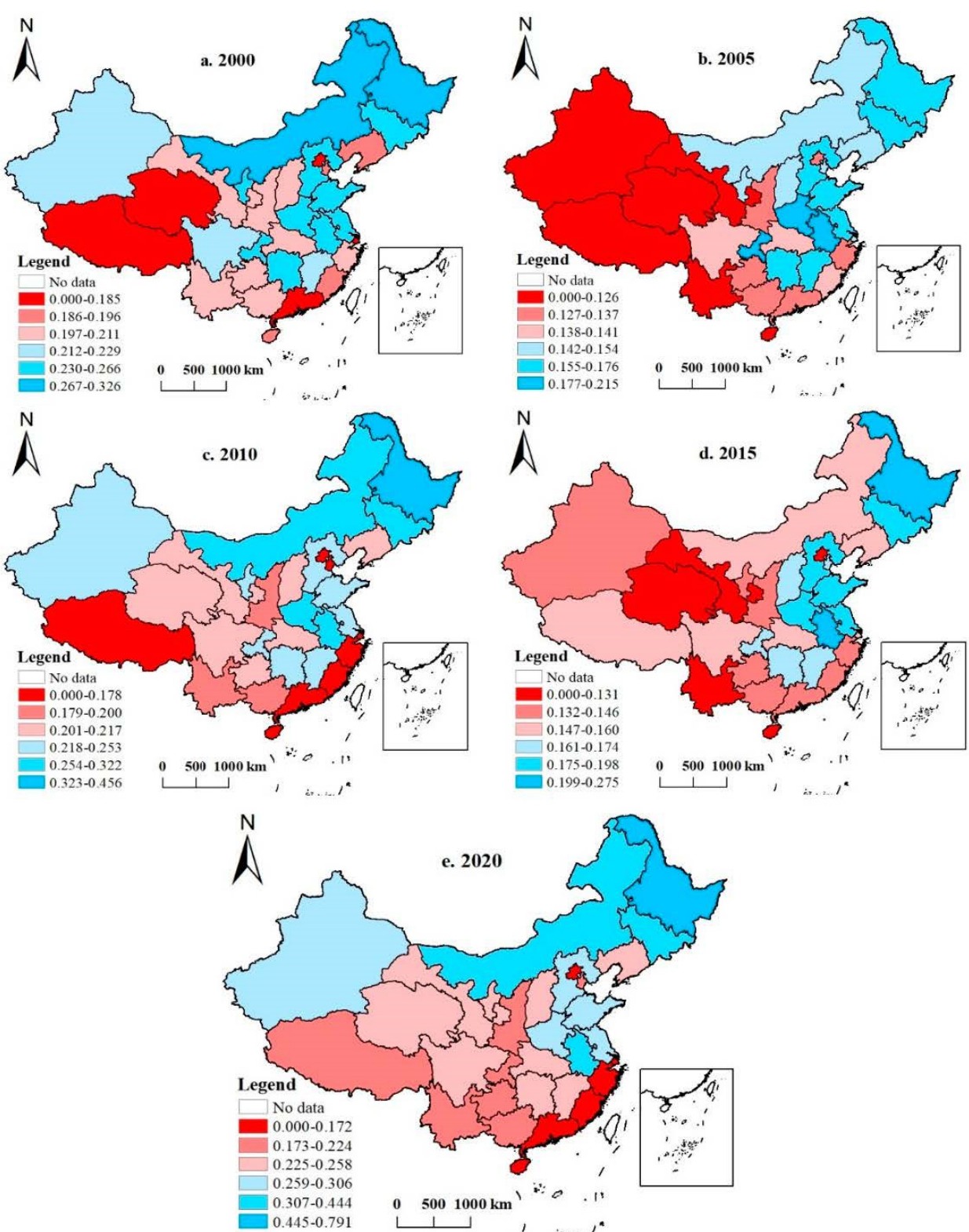

**Figure 3.** Evolution of spatial pattern of GTCL in China during 2000–2020.

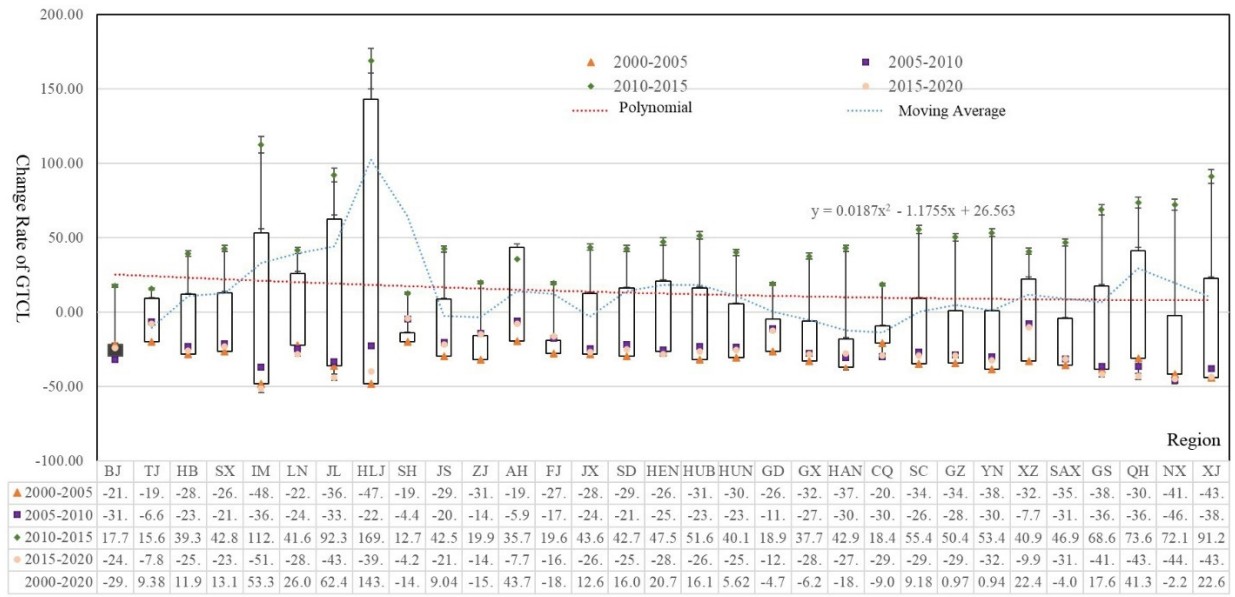

| | BJ | TJ | HB | SX | IM | LN | JL | HLJ | SH | JS | ZJ | AH | FJ | JX | SD | HEN | HUB | HUN | GD | GX | HAN | CQ | SC | GZ | YN | XZ | SAX | GS | QH | NX | XJ |
|---|---|---|---|---|---|---|---|---|---|---|---|---|---|---|---|---|---|---|---|---|---|---|---|---|---|---|---|---|---|---|---|
| ▲2000-2005 | -21. | -19. | -28. | -26. | -48. | -22. | -36. | -47. | -19. | -29. | -31. | -19. | -27. | -28. | -29. | -26. | -31. | -30. | -26. | -32. | -37. | -20. | -34. | -34. | -38. | -32. | -35. | -38. | -30. | -41. | -43. |
| ■2005-2010 | -31. | -6.6 | -23. | -21. | -36. | -24. | -33. | -22. | -4.4 | -20. | -14. | -5.9 | -17. | -24. | -21. | -25. | -23. | -23. | -11. | -27. | -30. | -30. | -26. | -28. | -30. | -7.7 | -31. | -36. | -36. | -46. | -38. |
| ♦2010-2015 | 17.7 | 15.6 | 39.3 | 42.8 | 112. | 41.6 | 92.3 | 169. | 12.7 | 42.5 | 19.9 | 35.7 | 19.6 | 43.6 | 42.7 | 47.5 | 51.6 | 40.1 | 18.9 | 37.7 | 42.9 | 18.4 | 55.4 | 50.4 | 53.4 | 40.9 | 46.9 | 68.6 | 73.6 | 72.1 | 91.2 |
| ●2015-2020 | -24. | -7.8 | -25. | -23. | -51. | -28. | -43. | -39. | -4.2 | -21. | -14. | -7.7 | -16. | -26. | -25. | -28. | -26. | -25. | -12. | -28. | -27. | -29. | -29. | -29. | -32. | -9.9 | -31. | -41. | -43. | -44. | -43. |
| 2000-2020 | -29. | 9.38 | 11.9 | 13.1 | 53.3 | 26.0 | 62.4 | 143. | -14. | 9.04 | -15. | 43.7 | -18. | 12.6 | 16.0 | 20.7 | 16.1 | 5.62 | -4.7 | -6.2 | -18. | -9.0 | 9.18 | 0.97 | 0.94 | 22.4 | -4.0 | 17.6 | 41.3 | -2.2 | 22.6 |

**Figure 4.** Change rate of GTCL in China during 2000–2020.

### 3.1.2. Global Characteristics of the Evolution of Spatial Pattern

Using the trend analysis tools in ArcGIS 10.2 software, this paper makes a three-dimensional intervisibility analysis on the overall trend of the GTCL in 31 provinces, municipalities and autonomous regions of China from 2000 to 2020 as research units. Taking the GTCL as the Z axis and the X and Y axes as the due east and due north directions, respectively, the spatial visualization results are obtained (Figure 5). The results show that there are significant spatial differences in the distribution of GTCL in China from 2000 to 2020. The overall distribution of cultivated land is basically the same from east to west, high in the north and low in the south.

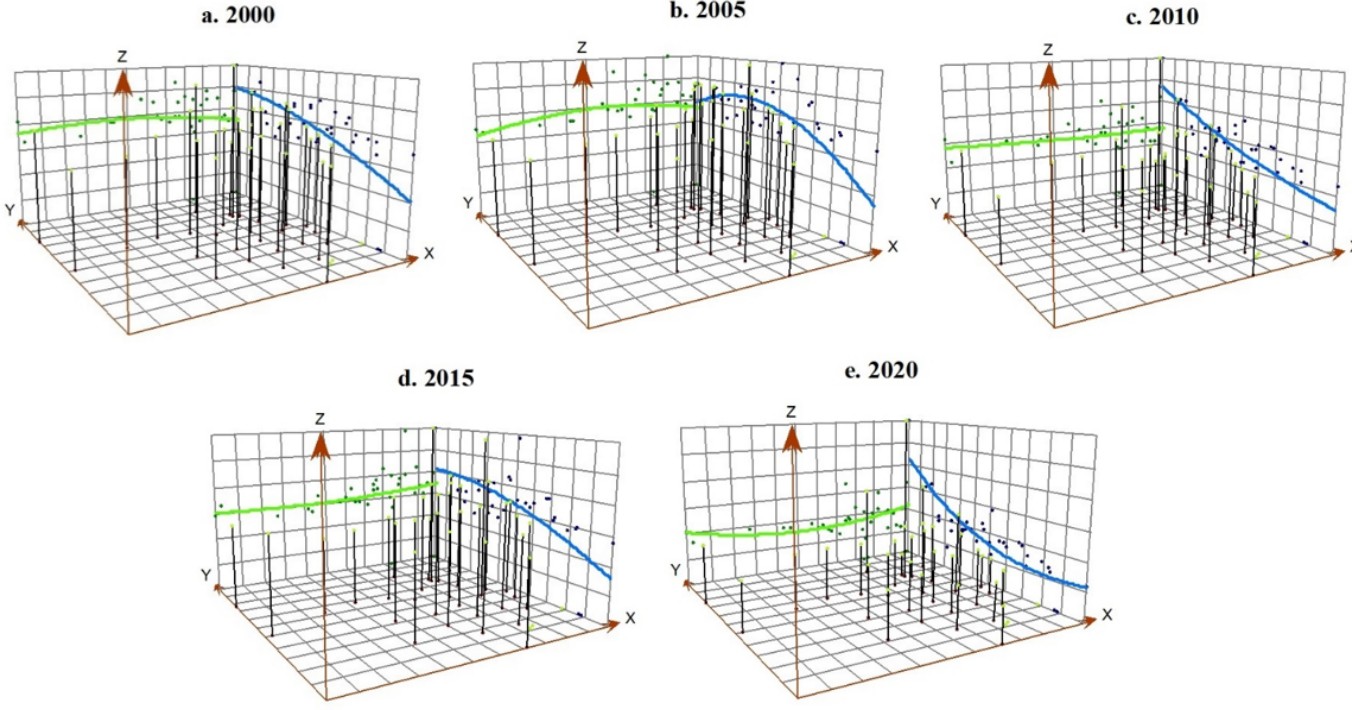

**Figure 5.** Trend analysis of GTCL in China during 2000–2020.

In GeoDa, a global spatial autocorrelation analysis was conducted for GTCL using Rook's criterion to calculate the Global Moran's I index (Figure 6).

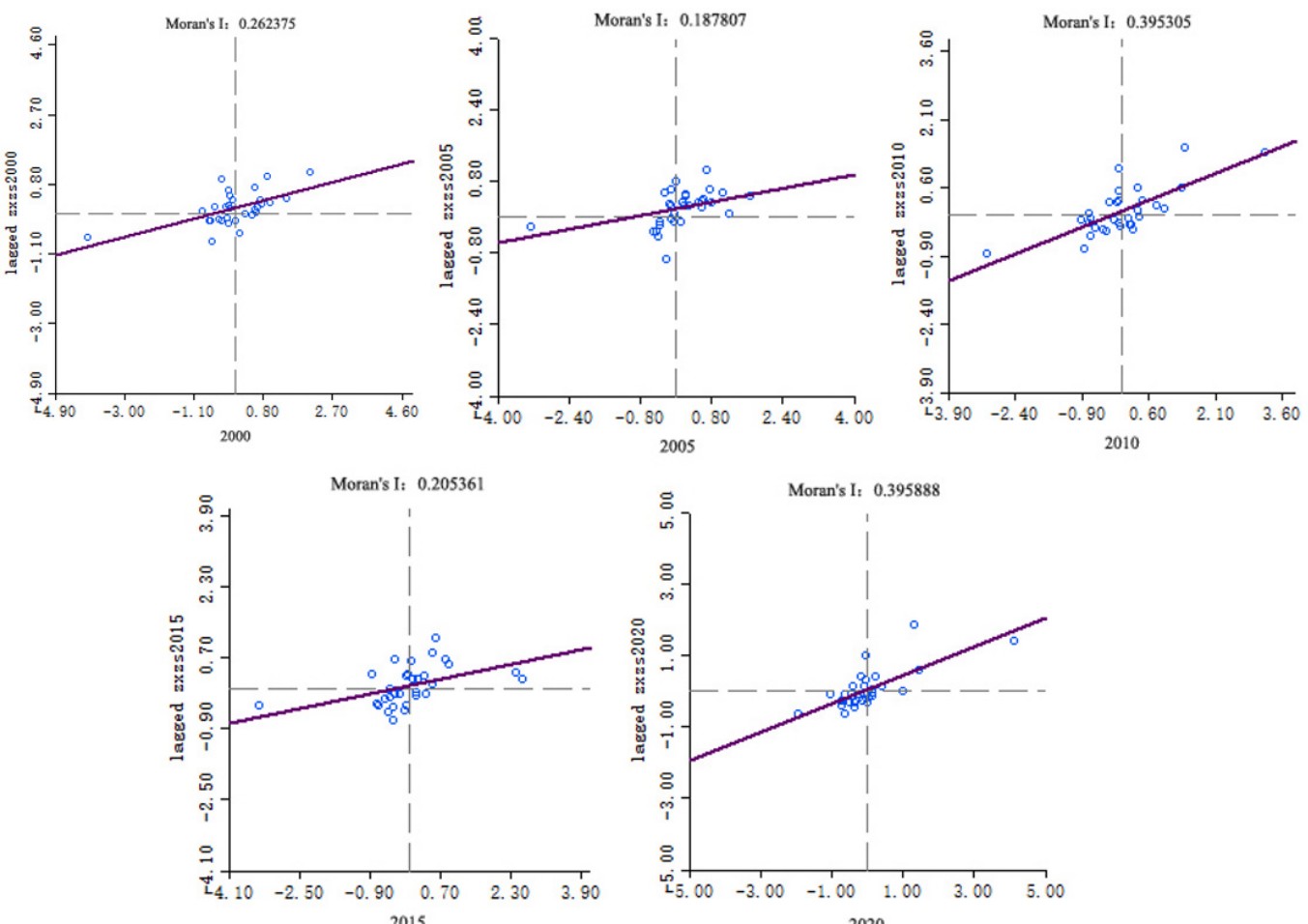

**Figure 6.** Moran's *I* index value of GTCL.

The global autocorrelation coefficients of Moran's I were all positive: 0.262, 0.188, 0.395, 0.205, and 0.396 from 2000 to 2020, respectively, and were divided into three categories according to the relative magnitude of each year: strong (absolute value $\geq$ 0.5), relatively weak (0.3 $\leq$ absolute value < 0.5), and weak (0 $\leq$ absolute value < 0.3). Overall, GTCL in each province, autonomous region and municipality showed a significant positive correlation between 2000 and 2020, and there were obvious regional clustering characteristics in space. From 2000 to 2005, the Moran's I value decreased, indicating a weak clustering distribution pattern among provinces, municipalities and autonomous regions. Compared with 2005, the Moran's I value increased significantly in 2010, which showed a weak clustering distribution pattern among provinces, municipalities and autonomous regions. Compared with 2015, Moran's I value in 2020 showed a significant increase, but it was still larger than that in 2010, indicating that the correlation of spatial distribution of GTCL among provinces, municipalities and autonomous regions increased in the period of 2015–2020, but the overall spatial differences showed an increasing trend.

### 3.2. Spatial and Temporal Patterns of "Water, Land, Food and Carbon" Changes

The spatial and temporal patterns of "water, land, food, and carbon" changes in each province, autonomous region and municipality were analyzed based on the evaluation indexes W (w), L (l), F (f), and C (c) (Figures 7 and 8).

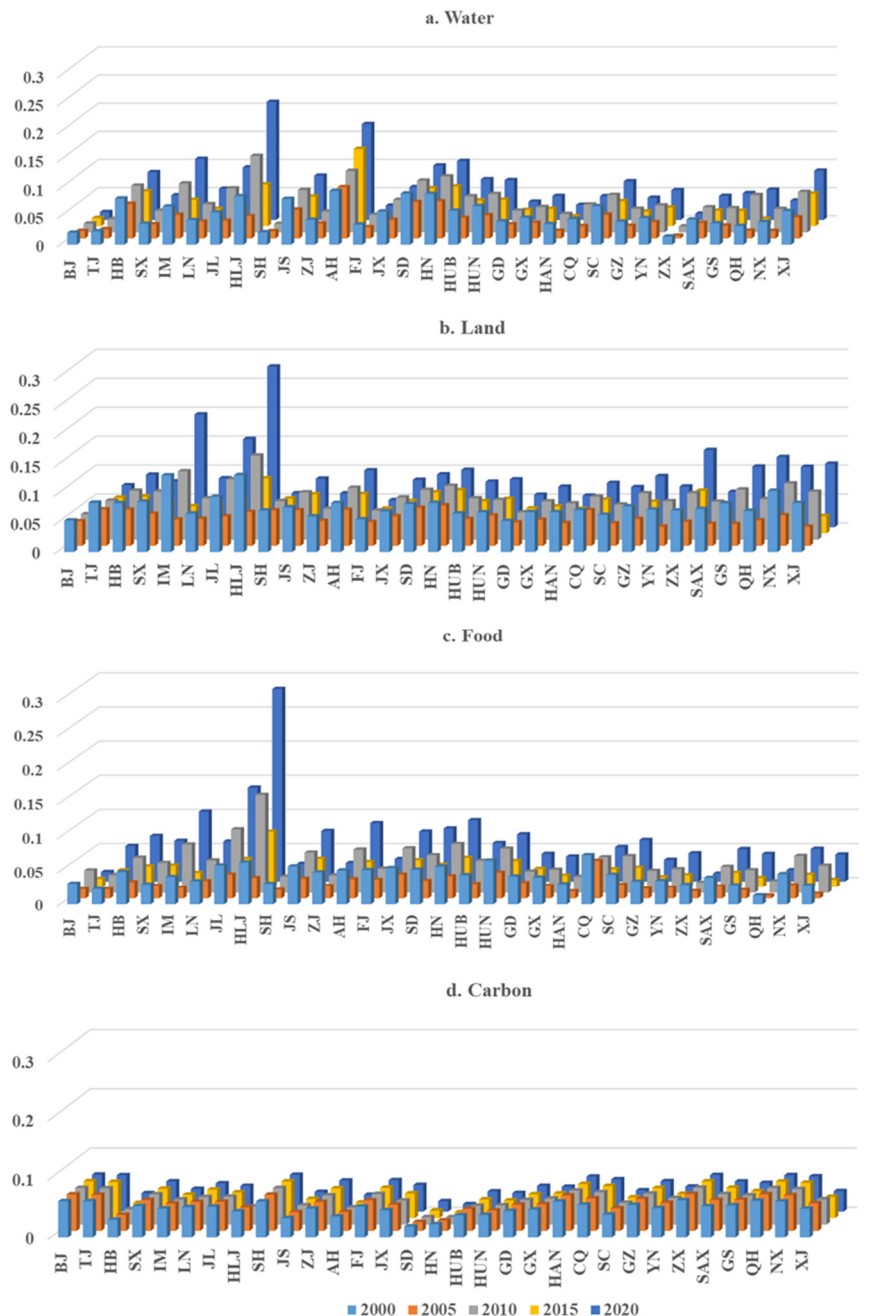

**Figure 7.** Temporal and spatial distribution of "Water-Land -Food-Carbon".

The water system W (w) evaluation index ranges from 0.0048 to 0.210, with a large fluctuation trend, mainly influenced by the virtual water self-sufficiency rate, effective irrigation area, water consumption per unit area of grain yield and water consumption per unit of cultivated land (Figure 8a). From 2000 to 2005, the highest rates of change in water were in Tibet, Ningxia and Hainan, with −66.82%, −66.37% and −62.34%, respectively, and the lowest was in Anhui, with a reduction of 4.44%, probably related to the reduced contribution of local water resources to the population's food consumption. From 2005 to 2010, the highest rate of water change was in Northwest China, with the largest increases in Qinghai and Ningxia, followed by Heilongjiang and Jilin in Northeast China, and the lowest was in Beijing, with an increase of 16.17%, probably related to the increase in the effective irrigated area. From 2010 to 2015, the trend was decreasing, with the highest rate of change in Qinghai and Ningxia, at 80.54% and 63.05%, followed by Jilin, at −56.63%, and Beijing, at −7.33%. From 2015 to 2020, the largest increases were in Qinghai, Heilongjiang

and Inner Mongolia, at 318.96%, 183.63% and 132.31%, with the smallest decrease in Beijing, at 1.29%, which is related to the increase of water consumption per unit of cultivated land.

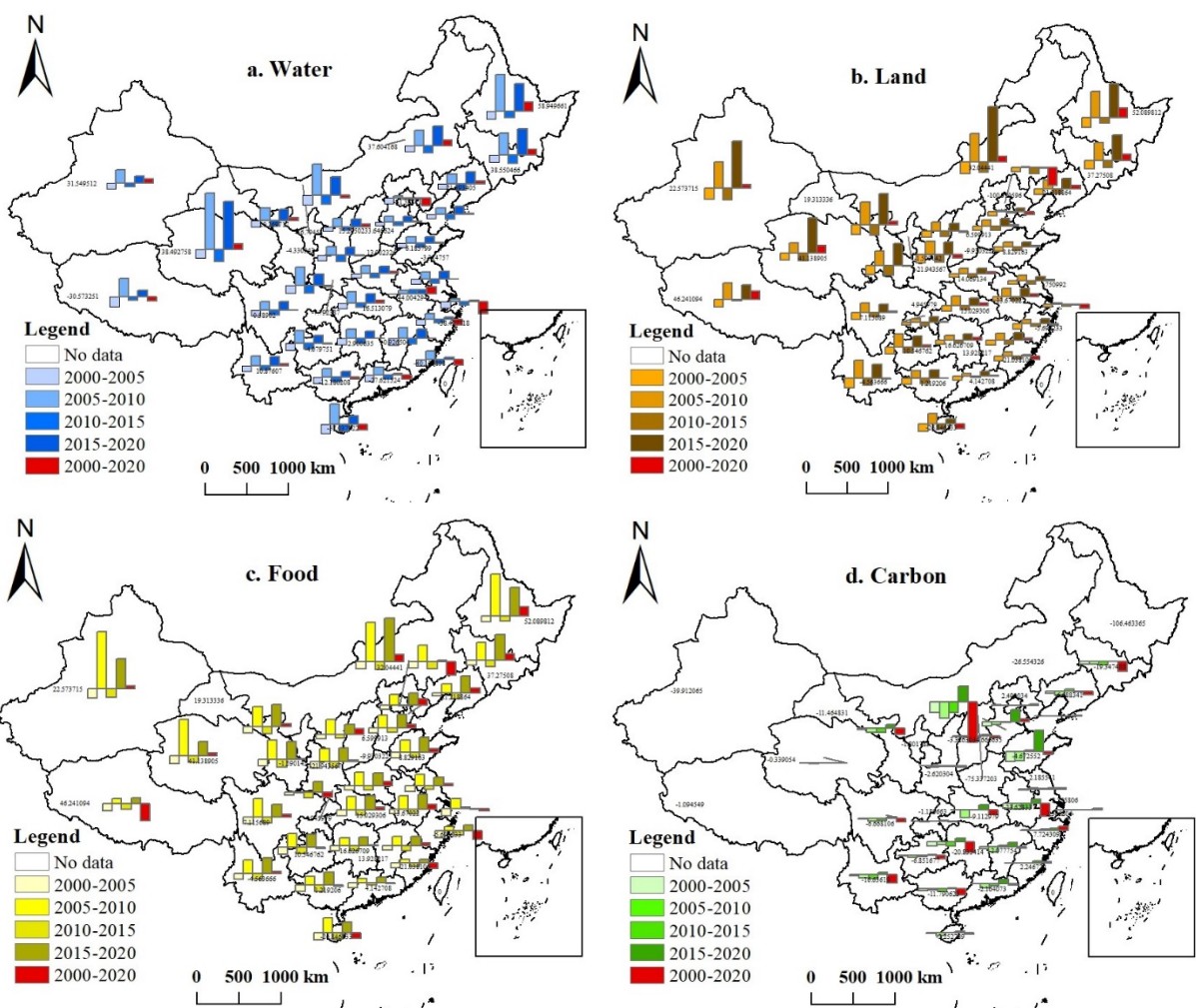

**Figure 8.** Evolution of spatial pattern of "Water–Land–Food–Carbon".

The L (l) evaluation index of the cultivated land system ranges from 0.027 to 0.278, showing a "W"-shaped variation trend, but with small fluctuations (Figure 8b). Moreover, 2000–2005 is the period of decreasing fluctuation, with an average annual change rate of 7.08%, which is related to the decrease of per capita cultivated land area; 2005–2010 is the period of increasing fluctuation, which is related to the increase of multiple crop index, among which the more typical ones are Inner Mongolia, Xinjiang and Heilongjiang, where the multiple crop index increased from 0.758, 0.936 and 0.857 to 0.980, 1.154 and 1.028; the fluctuating-decreasing phase from 2010 to 2015 is related to the increase of land reclamation rate. For example, the land reclamation rate in Shandong, Henan, Jiangsu and Anhui increased from 0.478, 0.475, 0.464 and 0.410 to 0.482, 0.485, 0.427, and 0.420, respectively; in the fluctuation-increasing phase from 2015 to 2020, Inner Mongolia, Xinjiang, Heilongjiang, and Gansu had the largest change rates, with average annual growth rates of 62.86%, 53.52%, 38.44%, and 35.37%, respectively, while Beijing, Shanghai, and Fujian had the smallest change rates. This is related to the per capita cultivated land area and the intensity of non-point source pollution.

The evaluation index of food system F (f) ranged from 0.0042 to 0.282, with a fluctuation growth trend and a more stable fluctuation trend (Figure 8c). The period of 2000–2005 was a decreasing phase, which was closely related to the instability of grain sown area and grain yield; the period of 2005–2010 was a growing phase, especially in Xinjiang,

Heilongjiang, Inner Mongolia and Qinghai with annual average growth rates of 99.63%, 72.79%, 68.57% and 63.53%, respectively, mainly influenced by the per capita grain yield, the proportion of sown area of grain crops and the ratio of food crops to cash crops. From 2010–2015 was a fluctuation-decreasing phase, with a relatively stable decrease, averaging 10% per year; 2015–2020 was a fluctuation-increasing trend, mainly influenced by the average grain yield.

The evaluation index of carbon system C (c) ranged from 0.0098 to 0.063, with less fluctuation than the values of the water, cultivated land and food systems, and the increasing trend of the evaluation index was not obvious, basically in a slow growth state, with a mean value of 0.045, which was probably related to the slow increase of carbon emissions from pesticides, fertilizers, agricultural films, tillage, total power of agricultural machinery and irrigation during the use of cultivated land (Figure 8d).

### 3.3. Water-Land-Food-Carbon" Coupling Coordination Analysis

From Figures 9 and 10, the coupling coordination degree of "water-land-food-carbon" of all provinces, municipalities and autonomous regions in China from 2000 to 2020 showed a trend of decreasing before increasing, and the lag of "water-land-food-carbon" system was improved. The coupling coordination in Northeast China first decreased and then increased, among which, the coupling coordination in Heilongjiang was higher than the other two provinces and changed from moderate coordination (0.55) to good coordination (0.74) with a rate of 35.15%; Jilin changed from low coordination (0.50) to good coordination (0.62); Liaoning changed between low coordination and reluctant coordination (0.44→0.38→0.46→0.38→0.49). In North China, Beijing's coupling coordination changed more steadily, all in a low coordination state; Inner Mongolia was between low and reluctant coordination (0.44→0.38→0.46→0.38→0.49) during 2000–2015, and the coupling coordination was higher than the other four provinces and municipalities in 2020, reaching a good coordination state (0.62). Anhui in East China changed from low coordination (0.50) to good coordination (0.57). Except for Anhui, the areas of Jiangsu, Zhejiang, Jiangxi and Shandong increased on the whole and were basically between low coordination and reluctant coordination, while Shanghai and Fujian decreased on the whole and the coupling coordination changed from low coordination (0.41, 0.44) to reluctant coordination (0.36, 0.39). Henan, Hubei and Hunan in Central China had less changes and were basically in the state of low coordination (0.48, 0.48, 0.49). Guangdong, Guangxi and Hainan in South China developed from low coordination (0.42, 0.45, 0.43) to reluctant coordination (0.36, 0.36, 0.31) from 2000 to 2005; from 2005 to 2020, Guangdong and Guangxi developed into low coordination (0.41, 0.43), and Hainan was still in reluctant coordination, but with a relatively large increase of 11.42%. Shaanxi, Gansu, Ningxia and Xinjiang in Northwest China all changed from low coordination (0.45, 0.43, 0.48, 0.45) to reluctant coordination, and then to low coordination before increasing to low coordination (0.45, 0.47, 0.48, 0.49); Qinghai changes were more volatile, developing from reluctant coordination (0.39) to verge of disorder (0.28, 0.30) before rising to low coordination (0.45). The coupling coordination of Tibet in Southwest China was relatively low, developing from reluctant coordination (0.39) to verge of disorder (0.28) from 2000 to 2005, changing to reluctant coordination (0.36) from 2005 to 2020, and showing an overall decreasing trend (0.39→0.36); Ningxia and Shaanxi showed a decreasing trend but were basically in a low coordination state; Gansu, Qinghai and Yunnan were in low coordination. The "water-land-food-carbon" system showed lagging development.

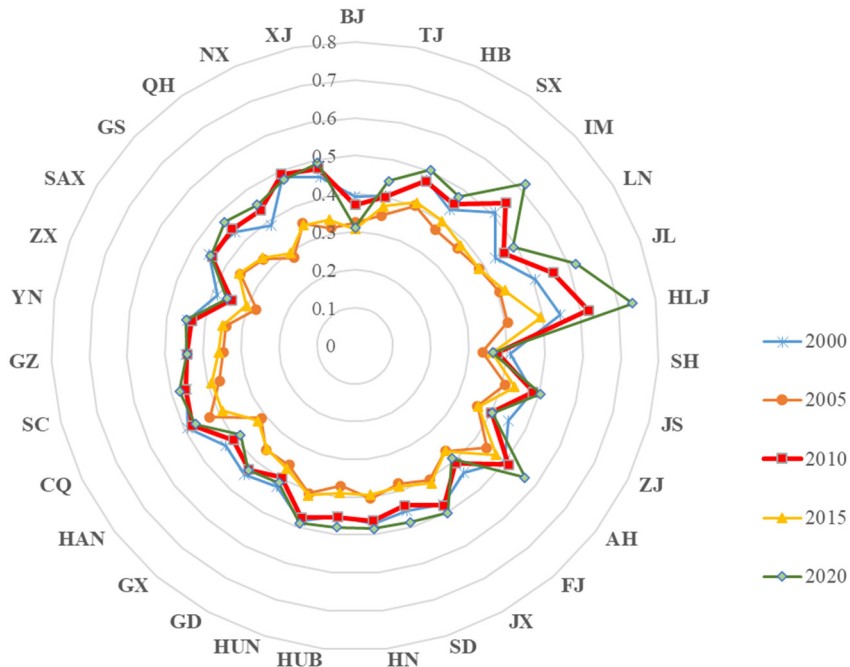

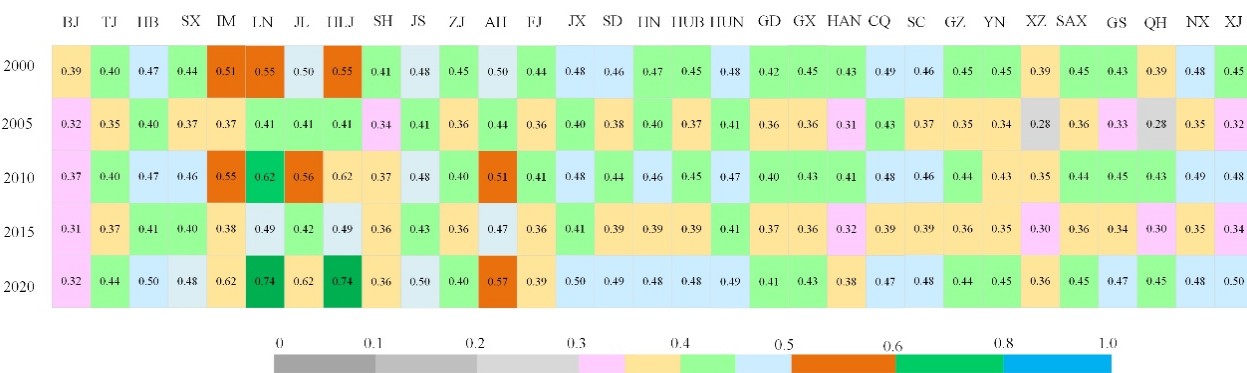

**Figure 9.** "Water–Land–Food–Carbon" coupling coordination degree. Note: BJ, TJ, HB, SX, IM, LN, JL, HLJ, SH, JS, ZJ, AH, FJ, JX, SD, HEN, HUB, HUN, GD, GX, HAN, CQ, SC, GZ, YN, ZX, SAX, GS, QH, NX, XJ is the abbreviation of Beijing, Tianjin, Hebei, Shanxi, Inner Mongolia, Liaoning, Jilin, Heilongjiang, Shanghai, Jiangsu, Zhejiang, Anhui, Fujian, Jiangxi, Shandong, Henan, Hubei, Hunan, Guangdong, Guangxi, Hainan, Chongqing, Sichuan, Guizhou, Yunnan, Tibet, Shannxi, Gansu, Qinghai, Ningxia, Xinjiang.

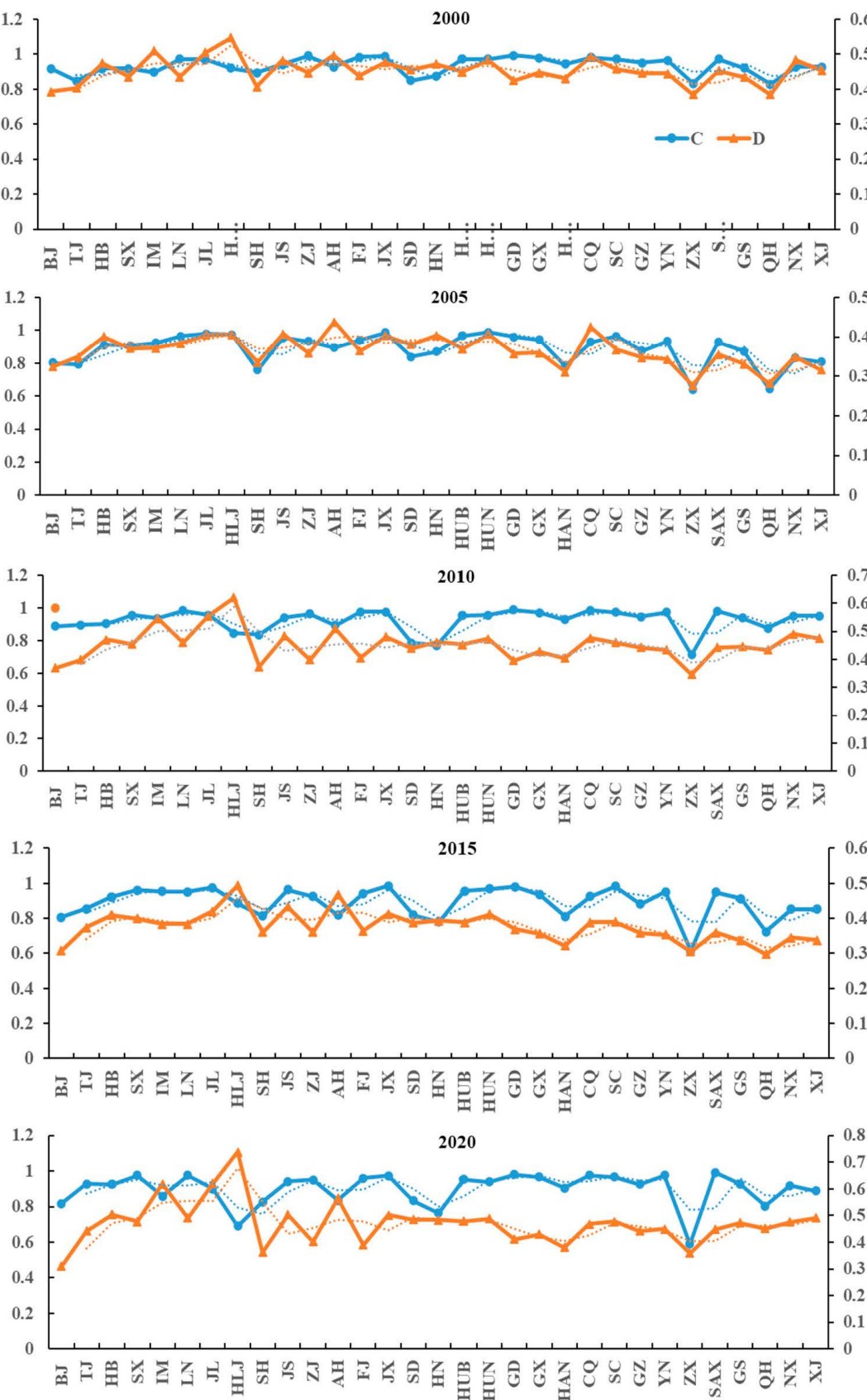

**Figure 10.** Development trend of coupling degree and coupling coordination degree of "Water–Land–Food–Carbon".

## 4. Discussion

### 4.1. Theoretical Implications of GTCL

At present, the Chinese government aims to promote green development and the overall green transition of economic and social development. The purpose of this paper is to realize a green, low-carbon, efficient and intensive spatial pattern of sustainable development of cultivated land-use based on the coupling coordination effect of "water-land-food-carbon" (Figure 2). From the perspective of a geographical pattern, the cultivated land-use system is not only a basic material space for economic and social development, but also an important part of rural habitat. In the process of promoting GTCL, priority shall be given to complex adaptive relationships between the whole and the local (element sharing and system coupling), conservation and development (sustainable use and food security), and equity and well-being (intergenerational balance and ecological welfare). For this reason, there is an urgent need to conduct theoretical and practical research on GTCL from the perspective of resource-factor linkage. GTCL is an inherent requirement for driving regional high-quality development, which refers to the sustainable use of cultivated land gradually shifting away from the dependence on high factor consumption and agro-ecological damage to a green development approach in which economic growth and resource conservation, carbon emission reduction and farmland ecosystem improvement are reinforcing each other. GTCL is not only the essential requirement of the natural properties of land, but also the inevitable requirement of sustainable utilization of resources and ecological civilization construction, as well as the value requirement of realizing the green transition of economic and social development in an all-round way. To promote GTCL, attention must be paid to the conceptual innovation from "independent" means of production to "systematic" resource community, as well as the transition from extensive use and excessive mode to a large-scale, intensive and sustainable way, thus moving towards the unification of value goals from farmers' personal interests to social public interests. Thus, there is a recognition that exploring the WLFC nexus under this framework (Figure 2) is a crucial first step toward effectively implementing its emerging policies for GDCL. For example, policymakers can better formulate legally binding development and protection objectives by analyzing the interaction between WLFC subsystems, and promote the combination of relevant policies and broader socio-economic activities in a timely manner, which will be beneficial to the common realization of the objectives of sustainable use of cultivated land, increase of food production, economic transformation, optimization of resources and environment, and mitigation of climate change.

### 4.2. Impact of GTCL on the WLFC Nexus

A compelling reason for analyzing GTCL from the "water-land-food-carbon" perspective is that the degree of water-soil matching, cultivated land reuse, grain yield, cultivated land-use and carbon emissions play a crucial role in achieving sustainable cultivated land-use. "Water-land-food-carbon" is the external condition that leads to changes in GTCL; GTCL is the internal basis that influences the synergy of "WLFC", and the external cause works through the internal cause. According to the analysis above, the development strategies in different periods have different emphasis on "WLFC". From 2000 to 2020, industrial pollution caused by the rapidly developing of industrialization had a negative impact on the resources and environment, the modern agricultural construction was started after the abolition of agricultural tax, and urban sprawl in the period of rapid urbanization led to the massive loss of cultivated land. After that, cultivated land protection policies were integrated into a larger natural resources management framework at national level, which improved the coupling coordination degree of WLFC. Furthermore, the in-depth adjustment of the global industrial chain, supply chain and value chain represented by grain trade has produced a transmission effect on many uncertain risks. However, the policy framework of cultivated land protection has been constantly improved, including regulations, policies, subsidies, standards, technical measures, etc. [49]. Overall, the "W"-shaped fluctuating uptrend of GTCL is observed. That is naturally why the average values of the comprehensive transition index exhibit decrease first and then increase with a

repeatedly periodic dynamic process. Other compelling reasons include the positive effects of the green transition in soil conservation, soil carbon sequestration, soil organic matter levels, water and nutrient retention, and biodiversity support, especially for farmland and pastures with low productivity or environmental sensitivity. In general, when the degree of soil-water matching is close to the critical range, green use plays a dominant role in controlling grain yield and carbon emissions, e.g., the Xinjiang Uygur Autonomous Region and the Inner Mongolia Autonomous Region in Northwest China (Figures 3e and 6); if GTCL is not properly regulated, then no matter how high the degree of land-water matching is, it cannot further increase grain yield or reduce carbon emissions, e.g., Zhejiang Province and Fujian Province in East China (Figure 8a,b and Figure 9). Since the coupling of "water-land-food-carbon" is complex and involves the overlapping combination of virtual resource flows and remote utilization patterns in the context of cross-food trade, the examination of the coupling coordination between the WLFC subsystems will help to deeply analyze the external effects of GTCL, mainly from two perspectives. First, the cultivated land-use system is a complex giant system with coupling interactions of single or multiple subsystems in the process of green transition, and its transition goal, transition direction, transition path and its internal mechanism are mainly governed by water, land, food and carbon. Second, whether it is the degree of water-land matching, cultivated land reuse and grain yield or carbon emission, they are all influencing factors in the process of GTCL. Based on the spatial and temporal patterns of GTCL and the results of "water-land-food-carbon" coupling coordination analysis, the influence of external factors can be fully reflected by monitoring data. Relying on the macroscopic analysis model established on the basis of the transition mechanism, the "details" of the process that are difficult to quantify can be greatly reduced, thus ensuring the scientific and reasonable research results.

### 4.3. Ductile Control Strategy Based on WLFC Nexus

In this paper, the K-Medians clustering function of GeoDa was applied to cluster the values of the coupling coordination degree of "water-land-food-carbon" of China's 31 provinces, municipalities and autonomous regions in 2020, to set the criteria of internal coordination development of cultivated land-use. Combined with the level of GTCL (Figures 3–5 and Figure 9), they were divided into five levels, I to V, namely: benefit leading area, quality improvement area, connotation tapping potential area, ductile development area and ecological reserve area (Table 4).

**Table 4.** Regulation type division.

| Category | | Region |
|---|---|---|
| I | Benefit leading area | HLJ, JL, LN, IM |
| II | Quality improvement area | HB, SD, HN, AH, JS, ZJ, JX, HUB, HUN |
| III | Connotation tapping potential area | SX, SAX, BJ, TJ, SH, CQ |
| IV | Ductile development area | GZ, SC, YN, GD, GX, FJ, HAN |
| V | Ecological reserve area | XZ, XJ, QH, GS, NX |

The benefit leading area (I) includes four regions, namely, Heilongjiang, Jilin, Liaoning and Inner Mongolia, whose cultivated land-use transition development and "water, land, food and carbon" coupling coordination were at a high level, especially in Heilongjiang, where the cultivated land-use transition index was as high as 0.79 in 2020, and the four subsystems were in good coordination (Figures 3, 9 and 10). Therefore, this type needs to give full play to the advantages of resource endowment and guide the cultivated land-use toward the efficient enhancement of ecological, production and living benefits, and promote farmers' income, efficient use of soil and water resources, conservation of agricultural biodiversity, and the maintenance and enhancement of soil fertility. Quality improvement area (II) includes Hebei, Shandong, Henan, Anhui, Jiangsu, Zhejiang, Jiangxi, Hubei and Hunan. The GTCL of these types of provinces was at a medium level, and the four subsystems were in low and moderate coordination (Figures 3, 9 and 10). This type of region was mostly located in the groundwater leakage area of the Huang-huai-hai Plain, and the

degradation and pollution of cultivated land due to agricultural production had obvious conflicts with ecological environmental protection and food safety guarantee. Therefore, it is urgent to introduce high-yielding and drought-tolerant food crops, promote the synergistic intensification of labor, capital, and technology to improve grain yield, and ensure food self-sufficiency based on agricultural science and technology. Connotation tapping potential area (III) includes Shanxi, Shaanxi, Beijing, Tianjin, Shanghai and Chongqing. This type of "water, land, food and carbon" coupling coordination was relatively low (Figures 9 and 10), basically in a state of reluctant coordination and low coordination, especially in Beijing, Tianjin and Shanghai, which had more carbon emissions from cultivated land-use but small grain yield. Therefore, special attention should be paid to the reduction of pesticide and chemical fertilizer use, and the implementation of carbon emission reduction and environmental pollution control. Ductile development area (IV) includes Guizhou, Sichuan, Yunnan, Guangdong, Guangxi, Fujian, and Hainan. The green development level of cultivated land-use in this type of region involved was low (Figure 4), with an average level of only 0.2, and the degree of coupling coordination of the four subsystems was also relatively low (Figures 9 and 10). Therefore, it's necessary to explore the development space of intensive use and grain yield potential in the region, and improve the ability of stable growth, efficient use, resistance to disturbance and resource support of cultivated land. In addition, farmers should be better motivated to expand grain sowing area and develop special agriculture and thus increase farmers' income. Ecological reserve area (V) includes Tibet, Xinjiang, Qinghai, Gansu and Ningxia. This type should improve the planting structure, cultivate as appropriate, pay attention to the protection of ecological functions such as water purification, $CO_2$ regulation, hydrological regulation, biodiversity protection, farming landscape, etc. to ensure food security and ecological safety.

## 5. Conclusions

The WLFC nexus is crucial to the sustainable development of cultivated land-use and human well-being. In this study, "water, land, food and carbon" is considered as the integration point for exploring GTCL. We constructed a theoretical analysis framework for GTCL based on the WLFC nexus and evaluated the spatial and temporal patterns of GTCL in all provinces (autonomous regions, municipalities) of China from 2000 to 2020. In fact, previous studies mainly focused on the impact of single factor management on cultivated land-use, but often ignored the key role of factor correlation at different levels. This study further determines WLFC nexus in combination with Chinese national conditions on the basis of WEF, and it helps policymakers to reduce the comprehensive cost in the decision-making process under the background of complex system management, which will provide basic support for GTCL research in China. However, due to the novelty of this analysis framework, the data set required for economic, social and ecological analysis has not yet been formed. Relevant statistics on the evolution of groundwater, the mechanism of diet nutrition, the effect of carbon source and sink have not been considered in the scope of this paper, which is also a deficiency of this research.

In addition, we analyzed the spatial and temporal characteristics of "water, land, food and carbon" and their coupling coordination. According to the results, GTCL, the changes of WLFC system and their coupling coordination degrees showed spatial and temporal coincidences with great consistency. From 2000 to 2020, GTCL index in each province (autonomous region, municipality) showed an overall "W"-shaped fluctuating uptrend, and the regional differences showed an overall increasing trend; we found that GTCL in each province (autonomous region, municipality) presented a significant positive correlation with obvious spatial characteristics of regional clustering; in the past five years, GTCL in Northeast China had a higher development level, followed by Central and North China, while South China was at a low level. In addition, the "water, land and food" system showed a more obvious "W"-shaped fluctuation, with the coupling coordination in Northeast China being higher and in good coordination while lower in East and Southwest China. To a certain extent, this trend promoted the green and low-carbon use of cultivated

land and the coordinated matching of water, land and food in China. Although there are substantial differences in cultivated land-use among regions, the common challenges drive the sharing and cooperation of similar policy approaches. Moreover, due to common scientific, technological and environmental issues, there will be great opportunities for cross-regional joint research and development and knowledge sharing in the future. It is concluded that this paper proposes a new solution to the GTCL based on theoretical and empirical analysis, which can promote the cooperation among public administration departments, business organizations and civil society when they develop or innovate these measures. Besides that, integrating and resolving the interests and responsibilities of different stakeholders from the institutional level, and taking into account the consistency of policy objectives and the common interests of technical requirements, it will enhance policy effectiveness.

Finally, the "water-land-food-carbon" effect of GTCL and the WLFC-based cultivated-land resilience control strategy were discussed. It should establish the planting system and green production system around green agricultural products and create a recycle agriculture model integrating planting and breeding; it also needs to organize a series of activities such as GTCL-related research for exhibition, public welfare training and digital communication to improve the understanding of stakeholders. At the same time, it is necessary to expand the scope of GTCL management according to the sustainable development standards and formulate the mechanism of GTCL protection and development strategy through the establishment of information sharing, data collection, risk warning and capacity-building, so as to better promote institutional, national and regional international cooperation. In general, this study proposed a comprehensive assessment system for GTCL, which may contribute to the sustainable use of cultivated land in the future. Nevertheless, the continuous and innovative research on GTCL requires a more comprehensive understanding of the interaction mechanism between ground and underground. We should not only pay attention to the internal material circulation in the whole food production chain, but also focus on the influencing mechanism of external factors. For example, how does the cultivated land protection policy affect GTCL? This will be of interest for future research.

**Author Contributions:** Conceptualization, S.N. and X.L.; methodology, S.N. and G.G.; software, S.N. and G.G.; formal analysis, X.L.; investigation, S.N.; data curation, S.N.; writing—original draft preparation, S.N. and X.L.; writing—review and editing, S.N. and X.L.; visualization, S.N. and G.G. All authors have read and agreed to the published version of the manuscript.

**Funding:** This paper won the support of the National Natural Science Foundation of China (Approval Number: 42071226). Fundamental Research Funds for the Central Universities (Grant/Award Number: N2214001, N2114006).

**Conflicts of Interest:** The authors declare no conflict of interest.

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
