# Peer review of "A New Framework of Green Transition of Cultivated Land-Use for the Coordination among the Water-Land-Food-Carbon Nexus in China"

_land, doi:10.3390/land11060933_

Round 1

Reviewer 1 Report

The reviewed article examined an interesting way a new framework of green transition of cultivated land-use for the coordination among water-land-food-carbon nexus in China. I believe the article is linguistically and grammatically correct. I rate the article at a high level. The literature is up-to-date and properly matched to the issues described in the text.

Author Response

Comment 1: The reviewed article examined an interesting way a new framework of green transition of cultivated land-use for the coordination among water-land-food-carbon nexus in China. I believe the article is linguistically and grammatically correct. I rate the article at a high level. The literature is up-to-date and properly matched to the issues described in the text.

Response 1: Thanks for your suggestion. According to your suggestion, we revised some details to make the paper more clear.

Reviewer 2 Report

Dear authors, please find my proposal for improving the manuscript.

Line 54:  excessive use of nitrogen and phosphorus fertilizers can't cause soil pollution. It can causes (leaching, erosion) and pollution of water resources. Its production can result in CO2 emissions. Please rephrase.

Line 55: "residual films can damage the soil structure of the tillage layer, causing difficult infiltration of groundwater and secondary soil salinization" Do you mean plastic films/mulch. Organic mulch is beneficial at all times. Please rephrase with plastic.

Line 56: "pesticide residues endanger human and animal health". Please rephrase this sentence. Endanger is a hard word. Pesticides present a risk if they are not used in a proper professional way. Use word risk.

Line 58: there is a need to find "sustainable" ways ... and protect the "environment in agroecosystems".

Line 64-65: " per capita cultivated land areas less than 1/2 of the world average" - please add the number in hectares per capita.

Restructure chapter 2. Theoretical framework and 3. Materials and methods  as follows

2.1. Case study and data source

2.2. Green Transition of Cultivated Land-use (GTCL)

2.3. Evaluation index system of GTCL

2.4. Determination of index weights

2.5. Evaluation method

2.6 Uncertainties and shortcomings (explain the weakness of the study and how could be overcome (money, time, better data, etc.)

Renumber chapter 4. Results to 3. Results

Line 350: What is meant by  W”-shaped fluctuating uptrend in relation to Figure 4.

Are boxes with bars presenting individual regions or years?

The X-axis and y-axis must have the title

Discussion:

It would be good to explain why a W-shaped trend is observed (economic, health crisis). So which internal (five-year plans - their goals vary) and external (global crisis-supply chains shifts) impact GTCL and WLFC.

Are there any inconsistencies in the legislation (development, population rise, rural area development, increase of GDB) that hinder the promotion of the green and low-carbon use of cultivated land and the coordinated matching of water, land and food in China

Conclusion:

Please answer in the text on these questions.

Why is this research unique?

What are the shortcomings/uncertainties of this research?

What did the scientific community learn out of it?

What are the benefits/recommendations for stakeholders (from the WLFC nexus)?

What are the recommendations for policymakers/legislators?

Future work?

Author Response

Referee: 2

Comment 1: Line 54: excessive use of nitrogen and phosphorus fertilizers can't cause soil pollution. It can causes (leaching, erosion) and pollution of water resources. Its production can result in CO2 emissions. Please rephrase.

Response 1: Thanks for your suggestion. According to your suggestion, we rewrote this part:

excessive use of nitrogen and phosphorus fertilizers can causes (leaching, erosion) and pollution of water resources, and its production can result in CO2 emissions.

Comment 2: Line 55: "residual films can damage the soil structure of the tillage layer, causing difficult infiltration of groundwater and secondary soil salinization" Do you mean plastic films/mulch. Organic mulch is beneficial at all times. Please rephrase with plastic.

Response 2: Thanks for your suggestion. According to your suggestion, we rewrote this part:

plastic residual films can damage the soil structure of the tillage layer, causing difficult infiltration of groundwater and secondary soil salinization.

Comment 3: Line 56: "pesticide residues endanger human and animal health". Please rephrase this sentence. Endanger is a hard word. Pesticides present a risk if they are not used in a proper professional way. Use word risk.

Response 3: Thanks for your comment. We reread the relevant references and rephrase this part:

pesticides present a risk if they are not used in a proper professional way, and even dam-age the ecological environment through the atmosphere, water bodies, soil, and food.

Comment 4: Line 58: there is a need to find "sustainable" ways ... and protect the "environment in agroecosystems".

Response 4: Thanks for your comment. According to your suggestion, we add the more appropriate word in this part:

Therefore, there is an urgent need to find sustainable ways to increase food supply and protect the environment in agroecosystems.

Comment 5: Line 64-65: " per capita cultivated land areas less than 1/2 of the world average" - please add the number in hectares per capita.

Response 5: Thanks for your comment. According to your suggestion, we rewrote this part:

The rapid development of urbanization has led to the loss of cultivated land, and the number in hectares per capita cultivated land areas less than 1/2 of the world average.

Comment 6: Restructure chapter 2. Theoretical framework and 3. Materials and methods  as follows

2.1. Case study and data source

2.2. Green Transition of Cultivated Land-use (GTCL)

2.3. Evaluation index system of GTCL

2.4. Determination of index weights

2.5. Evaluation method

2.6 Uncertainties and shortcomings (explain the weakness of the study and how could be overcome (money, time, better data, etc.)

Response 6: Thanks for your comment. According to your suggestion, we restructure chapter 2. Theoretical framework and 3. Materials and methods. Meanwhile, we add the 2.6 uncertainties and shortcomings in this part:

This paper preliminarily reveals the WLFC nexus in the evolution process of GTCL, and measures the coupling coordination degree between them on this basis. Due to the inconsistency of statistical caliber of data in different provinces, this current research has not been well studied the internal development of each subsystem of WLFC. However, the research data timeline for official statistics in this paper is selected according to the Five-Year Plan, which will be beneficial to provide a theoretical and empirical ground for formulation and implementation of cultivated land protection policy.

Comment 7: Renumber chapter 4. Results to 3. Results

Response 7: Thanks for your comment. According to your suggestion, we Renumber chapter 4. Results to 3. Results:

3.1. Spatial and temporal patterns of GTCL

3.1.1. Dynamic evolution characteristics of regional differences

3.1.2. Global characteristics of the evolution of spatial pattern

3.2. Spatial and temporal patterns of “water, land, food and carbon” changes

3.3. “Water-land-food-carbon” coupling coordination analysis

Comment 8: Line 350: What is meant by “W”-shaped fluctuating uptrend in relation to Figure 4. Are boxes with bars presenting individual regions or years?

The X-axis and y-axis must have the title

Response 8: Thanks for your comment. “W”-shaped indicates the average values of the comprehensive transition index in 2000, 2005, 2010, 2015 and 2020(0.202, 0.137, 0.206, 0.147 and 0.237)exhibits decrease first and then increase with a repeatedly periodic dynamic process.

These boxes with bars present individual regions’ Change rate of GTCL in China during 2000-2020(2000-2005, 2005-2010, 2010-2015, 2015-2020).

According to your suggestion, we add the corresponding title for Figure 4.

Comment 9: Study Area: Discussion:

It would be good to explain why a W-shaped trend is observed (economic, health crisis). So which internal (five-year plans - their goals vary) and external (global crisis-supply chains shifts) impact GTCL and WLFC.

Are there any inconsistencies in the legislation (development, population rise, rural area development, increase of GDB) that hinder the promotion of the green and low-carbon use of cultivated land and the coordinated matching of water, land and food in China

Response 9: Thanks for your comment. According to your suggestion, we rewrote this part:

4.1. Theoretical implications of GTCL

Thus there is a recognition that exploring WLFC nexus under this framework (Figure 2) is a crucial first step toward effectively implement its emerging policies for GDCL. For ex-ample, policymaker can better formulate legally binding development and protection objectives by analyzing the interaction between WLFC subsystems, and timely promote the combination of relevant policies and broader socio-economic activities, which will be beneficial to the common realization of the objectives of sustainable use of cultivated land, in-crease of food production, economic transformation, optimization of resources and environment, and mitigation of climate change.

4.2. Impact of GTCL on the WLFC nexus

According to the analysis above, the development strategies in different periods have different emphasis on “WLFC”. From 2000 to 2020, industrial pollution caused by the rap-idly developing of industrialization had a negative impact on the resources and environment, then the modern agricultural construction was started after the abolition of agricultural tax, urban sprawl in the period of rapid urbanization leads to the massive loss of cultivated land. After that, cultivated land protection policies is integrated into a larger natural resources management framework at national level, which will improve the coupling coordination degree of WLFC. Furthermore, the in-depth adjustment of global industrial chain, supply chain and value chain represented by grain trade has produced a transmission effect on many uncertain risks. However, the policy framework of cultivated land protection has been constantly improved, including regulations, policies, subsidies, standards, technical measures, etc. Overall, the “W”-shaped fluctuating uptrend of GTCL is observed. That is naturally why the average values of the comprehensive transition index exhibits decrease first and then increase with a repeatedly periodic dynamic process.

Comment 10: Conclusion:

Please answer in the text on these questions.

Why is this research unique?

Response 10: Thanks for your comment. According to your suggestion, we rewrote this part:

In fact, previous studies mainly focused on the impact of single factor management on cultivated land use, but often ignored the key role of factor correlation at different levels. This study further determines WLFC nexus in combination with Chinese national conditions on the basis of WEF, it helps policymakers to reduce the comprehensive cost in the decision-making process under the background of complex system management, which will provide basic support for GTCL research in China.

What are the shortcomings/uncertainties of this research?

Response 10: Thanks for your comment. According to your suggestion, we rewrote this part:

However, due to the novelty of this analysis framework, the data sets required for eco-nomic, social and ecological analysis has not yet been formed. Relevant statistics on the evolution of groundwater, the mechanism of diet nutrition, the effect of carbon source and sink have not been considered in the scope of this paper, it has also become deficiencies of the research.

What did the scientific community learn out of it?

Response 10: Thanks for your comment. According to your suggestion, we rewrote this part:

Although there are substantial differences in cultivated land use among regions, the common challenges drive the sharing and cooperation of similar policy approaches. Moreover, due to common scientific, technological and environmental issues, there will be great opportunities for cross regional joint research and development and knowledge sharing in the future. It is concluded that this paper proposes a new solution to the GTCL based on theoretical and empirical analysis, which can promote the cooperation among public administration departments, business organizations and civil society when they develop or innovate these measures.

What are the benefits/recommendations for stakeholders (from the WLFC nexus)?

Response 10: Thanks for your comment. According to your suggestion, we rewrote this part:

Besides that, integrating and resolving the interests and responsibilities of different stake-holders from the institutional level, and taking into account the consistency of policy objectives and the common interests of technical requirements, it will enhance policy effectiveness.

What are the recommendations for policymakers/legislators?

Response 10: Thanks for your comment. According to your suggestion, we rewrote this part:

It should establish the planting system and green production system around green agricultural products, and create a recycle agriculture model integrating planting and breeding; it also need to organize a series of activities such as GTCL related research to exhibition, public welfare training and digital communication to improve the understanding of stakeholders. At the same time, it is necessary to expand the scope of GTCL management according to the sustainable development standards, and formulate the mechanism of GTCL protection and development strategy through the establishment of information sharing, data collection, risk warning and capacity-building, so as to better promote institutional, national and regional international cooperation.

Future work?

Response 10: Thanks for your comment. According to your suggestion, we rewrote this part:

Nevertheless, the continuous and innovative research on GTCL requires a more comprehensive understanding of the interaction mechanism between ground and underground. We should not only pay attention to the internal material circulation in the whole food production chain, but also focus on the influencing mechanism of external factors. For example, how does the cultivated land protection policy affect GTCL? It will be the next interests for future research.

Reviewer 3 Report

In the present context, in which ecological problems are increasing, there is a growing concern to find strategies for sustainable development in the rural settings that make it possible to preserve ecological resources and natural environments for future generations, and at the same time to cultivate efficiently in order to achieve the necessary increases in food production in situations of constant population growth. In these circumstances, articles such as this one, are very timely, and can therefore be said to deal with a topic of great interest today. The text uses clear and precise language so it is easily understood. In addition, its reasoning is clearly stated and well sequenced. All of this makes it not only easy to read, but also attractive and pleasant as one progresses through the text. Apart from all of the above, another strength of the manuscript is that it is based on robust empirical research, which is solidly grounded from a methodological point of view.

My only recommendation is that the Conclusions should be somewhat longer and more detailed than they are in the present version. Moreover, these conclusions should not focus so much on explaining the results by Chinese regions of the study conducted, which, moreover, have already been described to a large extent in previous sections. Instead, perhaps it would be more appropriate to devote the conclusions to taking stock of the usefulness of the analytical model employed in the study, as well as to outlining possible extrapolations of this case study to other similar cases within and outside China.

Author Response

Comment 1: In the present context, in which ecological problems are increasing, there is a growing concern to find strategies for sustainable development in the rural settings that make it possible to preserve ecological resources and natural environments for future generations, and at the same time to cultivate efficiently in order to achieve the necessary increases in food production in situations of constant population growth. In these circumstances, articles such as this one, are very timely, and can therefore be said to deal with a topic of great interest today. The text uses clear and precise language so it is easily understood. In addition, its reasoning is clearly stated and well sequenced. All of this makes it not only easy to read, but also attractive and pleasant as one progresses through the text. Apart from all of the above, another strength of the manuscript is that it is based on robust empirical research, which is solidly grounded from a methodological point of view.

My only recommendation is that the Conclusions should be somewhat longer and more detailed than they are in the present version. Moreover, these conclusions should not focus so much on explaining the results by Chinese regions of the study conducted, which, moreover, have already been described to a large extent in previous sections. Instead, perhaps it would be more appropriate to devote the conclusions to taking stock of the usefulness of the analytical model employed in the study, as well as to outlining possible extrapolations of this case study to other similar cases within and outside China.

Response 1: Thanks for your suggestion. According to your suggestion, we rewrote this part:

The WLFC nexus is crucial to the sustainable development of cultivated land use and human well-being. In this study, “water, land, food and carbon” is considered as the integration point for exploring GTCL. We constructed a theoretical analysis framework for GTCL based on the WLFC nexus and evaluated the spatial and temporal patterns of GTCL in all provinces (autonomous regions, municipalities) of China from 2000 to 2020. In fact, previous studies mainly focused on the impact of single factor management on cultivated land use, but often ignored the key role of factor correlation at different levels. This study further determines WLFC nexus in combination with Chinese national conditions on the basis of WEF, it helps policymakers to reduce the comprehensive cost in the decision-making process under the background of complex system management, which will provide basic support for GTCL research in China. However, due to the novelty of this analysis framework, the data sets required for economic, social and ecological analysis has not yet been formed. Relevant statistics on the evolution of groundwater, the mechanism of diet nutrition, the effect of carbon source and sink have not been considered in the scope of this paper, it has also become deficiencies of the research.

In addition, we analyzed the spatial and temporal characteristics of “water, land, food and carbon” and their coupling coordination. According to the results, GTCL, the changes of WLFC system and their coupling coordination degrees showed spatial and temporal coincidences with great consistency. From 2000 to 2020, GTCL index in each province (autonomous region, municipality) showed an overall “W”-shaped fluctuating uptrend, and the regional differences showed an overall increasing trend; we found that GTCL in each province (autonomous region, municipality) presented a significant positive correlation with obvious spatial characteristics of regional clustering; in the past five years, GTCL in Northeast China had a higher development level, followed by Central and North China, while South China was at a low level. In addition, the “water, land and food” system showed a more obvious “W”-shaped fluctuation, with the coupling coordination in Northeast China being higher and in good coordination while lower in East and Southwest China. To a certain extent, this trend promoted the green and low-carbon use of cultivated land and the coordinated matching of water, land and food in China. Although there are substantial differences in cultivated land use among regions, the common challenges drive the sharing and cooperation of similar policy approaches. Moreover, due to common scientific, technological and environmental issues, there will be great opportunities for cross regional joint research and development and knowledge sharing in the future. It is concluded that this paper proposes a new solution to the GTCL based on theoretical and empirical analysis, which can promote the cooperation among public administration departments, business organizations and civil society when they develop or innovate these measures. Besides that, integrating and resolving the interests and responsibilities of different stakeholders from the institutional level, and taking into account the consistency of policy objectives and the common interests of technical requirements, it will enhance policy effectiveness.

Finally, the “water-land-food-carbon” effect of GTCL and the WLFC-based cultivated land resilience control strategy were discussed. It should establish the planting system and green production system around green agricultural products, and create a recycle agriculture model integrating planting and breeding; it also need to organize a series of activities such as GTCL related research to exhibition, public welfare training and digital communication to improve the understanding of stakeholders. At the same time, it is necessary to expand the scope of GTCL management according to the sustainable development standards, and formulate the mechanism of GTCL protection and development strategy through the establishment of information sharing, data collection, risk warning and capacity-building, so as to better promote institutional, national and regional international cooperation. In general, this study proposed a comprehensive assessment system for GTCL, which may contribute to the sustainable use of cultivated land in the future. Nevertheless, the continuous and innovative research on GTCL requires a more comprehensive understanding of the interaction mechanism between ground and underground. We should not only pay attention to the internal material circulation in the whole food production chain, but also focus on the influencing mechanism of external factors. For example, how does the cultivated land protection policy affect GTCL? It will be the next interests for future research.
